# The global forest above-ground biomass pool for 2010 estimated from high-resolution satellite observations

Maurizio Santoro[1], Oliver Cartus[1], Nuno Carvalhais[2,3], Danaë M. A. Rozendaal[4,5,6], Valerio Avitabile[7], Arnan Araza[4], Sytze de Bruin[4], Martin Herold[4], Shaun Quegan[8], Pedro Rodríguez-Veiga[9,10], Heiko Balzter[9,10], João Carreiras[8], Dmitry Schepaschenko[11,12,13], Mikhail Korets[14], Masanobu Shimada[15], Takuya Itoh[16], Álvaro Moreno Martínez[17,18], Jura Cavlovic[19], Roberto Cazzolla Gatti[20], Polyanna da Conceição Bispo[9,21], Nasheta Dewnath[22], Nicolas Labrière[23], Jingjing Liang[24], Jeremy Lindsell[25,26], Edward T.A. Mitchard[27], Alexandra Morel[28], Ana Maria Pacheco Pascagaza[9], Casey M. Ryan[27], Ferry Slik[29], Gaia Vaglio Laurin[30], Hans Verbeeck[31], Arief Wijaya[32], Simon Willcock[33]

[1]Gamma Remote Sensing, 3073 Gümligen, Switzerland

[2]Max Planck Institute for Biogeochemistry, Hans Knöll Strasse 10, 07745 Jena, Germany

[3]Departamento de Ciências e Engenharia do Ambiente, DCEA, Faculdade de Ciências e Tecnologia, FCT, Universidade Nova de Lisboa, 2829-516 Caparica, Portugal

[4]Laboratory of Geo-Information Science and Remote Sensing, Wageningen University & Research, Droevendaalsesteeg 3, 6708 PB Wageningen, The Netherlands

[5]Plant Production Systems Group, Wageningen University & Research, P.O. Box 430, 6700 AK, Wageningen, The Netherlands

[6]Centre for Crop Systems Analysis, Wageningen University & Research, P.O. Box 430, 6700 AK, Wageningen, The Netherlands

[7]European Commission, Joint Research Centre, Ispra, Italy

[8]National Centre for Earth Observation (NCEO), University of Sheffield, Sheffield, S3 7RH, United Kingdom

[9]Centre for Landscape and Climate Research, School of Geography, Geology and the Environment, University of Leicester, LE1 7RH, United Kingdom

[10]National Centre for Earth Observation (NCEO), Leicester, LE1 7RH, United Kingdom

[11]International Institute for Applied Systems Analysis, Schlossplatz 1, A-2361 Laxenburg, Austria

[12]Center of Forest Ecology and Productivity of the Russian Academy of Sciences, Profsoyuznaya 84/32/14, Moscow, 117997, Russia

[13]Institute of Ecology and Geography, Siberian Federal University, 660041, Krasnoyarsk, 79 Svobodny Prospect, Russia

[14]Laboratory of Ecophysiology of Permafrost Systems, V.N. Sukachev Institute of Forest of the Siberian Branch of Russian Academy of Sciences – separated department of the KSC SB RAS, Krasnoyarsk, 660036, Russia

[15]Tokyo Denki University, School of Science and Engineering, Division of Architectural, Civil and Environmental Engineering

[16]Remote Sensing Technology Center of Japan, Tokyu Reit Toranomon Bldg, 3f, 3-17-1 Toranomon, Minato-Ku, Tokyo, 105-0001, Japan

[17]Image Processing Laboratory (IPL), Universitat de València, València, Spain

[18]Numerical Terradynamic Simulation Group (NTSG), University of Montana, Missoula, USA

[19]University of Zagreb, Faculty of Forestry and Wood Technology, Department of Forest Inventory and Management, Svetosimunska cesta 23, 10000, Zagreb, Croatia

[20]Biological Institute, Tomsk State University, 634050 Tomsk, Russia

[21]School of Environment, Education and Development/Department of Geography, University of Manchester, Oxford Road, M13 9PL Manchester, United Kingdom

[22]Guyana Forestry Commission, 1 Water Street, Kingston, Georgetown, Guyana

[23]Laboratoire Évolution et Diversité Biologique, UMR 5174 (CNRS/IRD/UPS), 31062 Toulouse Cedex 9, France

[24]Department of Forestry and Natural Resources, Purdue University

[25]A Rocha International, Cambridge, United Kingdom
[26]The RSPB Centre for Conservation Science, Bedfordshire, United Kingdom
[27]University of Edinburgh, School of GeoSciences, Crew Building, The King's Buildings, Edinburgh, EH9 3FF, United Kingdom
[28]Department of Geography and Environmental Sciences, University of Dundee, United Kingdom
[29]Faculty of Science,University Brunei Darussalam, Jln Tungku Link, Gadong, BE1410, Brunei Darussalam
amma Remote Sensing, 3073 Gümligen, Switzerland
[30]Department for Innovation in Biological, Agro-Food and Forest Systems (DIBAF), University of Tuscia, 01100 Viterbo, Italy
[31]CAVElab – Computational and Applied Vegetation Ecology, Department of Environment, Ghent University, Coupure Links
653, 9000 Gent, Belgium
[32]World Resources Institute Indonesia (WRI Indonesia), Department of Research, Data & Innovation, Wisma PMI, 3rd Floor, Jl. Wijaya I/63, Kebayoran Baru, South Jakarta, Indonesia
[33]School of Natural Sciences, Bangor University, United Kingdom

*Correspondence to*: Maurizio Santoro (santoro@gamma-rs.ch)

**Abstract.** The terrestrial forest carbon pool is poorly quantified, in particular in regions with low forest inventory capacity. By combining multiple satellite observations of synthetic aperture radar (SAR) backscatter around the year 2010, we generated a global, spatially explicit dataset of above-ground forest biomass (dry mass, AGB) with a spatial resolution of 1 ha. Using an extensive database of 110,897 AGB measurements from field inventory plots, we show that the spatial patterns and magnitude of AGB are well captured in our map with the exception of regional uncertainties in high carbon stock forests with AGB > 250

Mg ha$^{-1}$ where the retrieval was effectively based on a single radar observation. With a total global AGB of 522 Pg, our estimate of the terrestrial biomass pool in forests is lower than most estimates published in the literature (426 - 571 Pg). Nonetheless, our dataset increases knowledge on the spatial distribution of AGB compared to the global Forest Resources Assessment (FRA) by the Food and Agriculture Organization (FAO) and highlights the impact of a country's national inventory capacity on the accuracy of the biomass statistics reported to the FRA. We also reassessed previous remote sensing

AGB maps, and identify major biases compared to inventory data, up to 120% of the inventory value in dry tropical forests, in the sub-tropics and temperate zone. Because of the high level of detail and the overall reliability of the AGB spatial patterns, our global dataset of AGB is likely to have significant impacts on climate, carbon and socio-economic modelling schemes, and provides a crucial baseline in future carbon stock changes estimates. The dataset is available at: https://doi.pangaea.de/10.1594/PANGAEA.894711 (Santoro, 2018).

**1 Introduction**

Above-ground live biomass (AGB) is identified as one of 54 Essential Climate Variables (ECVs) by the Global Climate Observing System (GCOS) because of its major role in the global carbon cycle. Biomass stores carbon removed from the atmosphere by photosynthesis in long-lived woody pools and yields to carbon emissions to the atmosphere when disturbed. Hence, accurate knowledge of its magnitude and spatial distribution is a key, and currently poorly constrained, part of the

carbon cycle (Houghton, 2005). Information on forest biomass is required to quantify forest resources and determine their

benefit in terms of ecosystem services (Schepaschenko et al., 2015; Reichstein and Carvalhais, 2019), climate change mitigation and biodiversity conservation (Soto-Navarro et al., 2020). Biomass estimates allow inferring emissions from forest degradation (Houghton et al., 2009; Li et al., 2017) and assisting with the derivation of emission factors (IPCC, 2006; Herold et al., 2019). Information on biomass also directly supports policy by quantifying national carbon stocks in the context of reducing emissions from deforestation and degradation (REDD+), the Paris Agreement on Climate Change and the United Nations Sustainable Development Goals (Gibbs et al., 2007; Herold et al., 2019). Finally, an improved knowledge on carbon stock patterns and dynamics from a better knowledge of forest biomass pools helps to constrain Earth System models (Carvalhais et al., 2014; Ciais et al., 2014; Bloom et al., 2016; Thurner et al., 2016; Baccini et al., 2017; Thum et al., 2017; Le Quéré et al., 2018; Exbrayat et al., 2019).

Previous estimates have suggested that plants store about 80% of the live biomass forming the Earth's biosphere, with an estimated carbon pool of 450 PgC (Bar-On et al., 2018). Around 320 PgC were allocated to the AGB, representing approximately 70% of the overall pool, most of it stored in woody biomass (Bar-On et al., 2018). However, our knowledge of the terrestrial woody biomass stock is relatively uncertain (Houghton et al., 2009). This uncertainty is well illustrated by the variance among forest biomass estimates from inventory data. A global assessments of biomass in forests for the year 2007 reported 362 PgC based on a compilation of forest inventory resources (Pan et al., 2011) whereas approximately 300 PgC were reported for the year 2010 based on the national contributions to the Food and Agriculture Organization (FAO) Forest Resources Assessment (FRA) (FAO, 2010). This uncertainty is a consequence of the uneven characterization of AGB in terms of precision and timeliness of measurements (Houghton et al., 2009; Ciais et al., 2014), and the lack of a universal inventory system using a standard set of survey and reporting procedures. Most countries in the temperate and boreal zones have National Forest Inventories (NFIs) that use systematic regular sampling, albeit some national differences e.g., in the definition of forest area (Tomppo et al., 2010). In contrast, many of the tropical countries have less developed inventory infrastructures or have only recently started to develop such infrastructure, often with the support of international initiatives (e.g., the UN REDD programme).

Remote sensing observations allow the estimation of global ecosystem properties and parameters (Schimel et al., 2015). No single measurement from remote sensing, however, represents a direct measure of the forest AGB. Nonetheless, the demand on spatially explicit estimates of AGB and the wide range of satellite observations collected in the last decades has fostered the development of a multitude of retrieval models based either on empirical regression techniques, physically-based mathematical models or machine learning algorithms (Lucas et al., 2015; Lu et al., 2016; Santoro and Cartus, 2018). The incapacity of remote sensing to measure biomass and the approximations in retrieval models cause inaccurate estimates of AGB at the pixel level. Even the spatial distribution of AGB in global and biome-specific maps of remotely sensed AGB (Kindermann et al., 2008; Saatchi et al., 2011b; Baccini et al., 2012; Thurner et al., 2014; Liu et al., 2015; Avitabile et al.,

2016; Hu et al., 2016) presents sometimes remarkable differences (Mitchard et al., 2013; Ometto et al., 2014; Schepaschenko
et al., 2015; Rodríguez-Veiga et al., 2017), implying a strong variability of the global biomass pool estimate (Table S1).

Global datasets of forest AGB from remote sensing observations represent the stocks for a snapshot ranging between 2000 and 2010, and their coarse spatial resolution ($\geq$ 500 m) hinders description of the fine-scale spatial variability of biomass. This aspect is of major importance when trying to capture changes in land use, natural disturbances and growth patterns (Houghton et al., 2009) or monitor management practices (Erb et al., 2018). Here, we assembled a wide set of publicly available RADAR, LiDAR and optical satellite observations suited to estimate forest variable with the objective of generating a high-resolution, global map of spatially explicit estimates of forest AGB (unit: Mg of dry mass per hectare) so to provide more recent, more detailed and possibly more accurate information on the spatial distribution of global AGB with respect to existing datasets. Our forest AGB map has a pixel size of 1 ha and is based on satellite remote sensing observations from around the year 2010. Here, we present the dataset together with an assessment of its validity using an extensive database of plot-level measurements of AGB. The significance of our map estimates is demonstrated in the context of biomass stock assessments by benchmarking with respect to the FAO FRA country statistics. In addition, we compare our estimates and other published estimates of forest AGB derived from remote sensing observations with AGB measurements from inventory plots to illustrate the reliability of our estimates.

## 2 Material and Methods

### 2.1 Satellite data

The spatially explicit estimates of AGB were based on the radar backscattered intensity recorded by the Phased Array-type L-band Synthetic Aperture Radar (PALSAR) instrument, onboard the Advanced Land Observing Satellite (ALOS) satellite, and the Advanced Synthetic Aperture Radar (ASAR) instrument operating at C-band, onboard the Environmental Satellite (Envisat) (Supplement Section A.1). In addition, LiDAR-based metrics and surface reflectances were used throughout the process of biomass estimation.

ALOS PALSAR was an active microwave sensor operating at L-band (wavelength of 23 cm). Between 2006 and 2011, PALSAR acquired images in the Fine Beam Dual-polarisation (FBD) mode with 20 m spatial resolution. Image acquisition followed a predefined observation scenario with the aim of achieving spatially and temporally consistent large-scale observational datasets (Rosenqvist et al., 2007). Summer-time acquisitions from the FBD mode (mostly May to October) were used by the Japan Aerospace Exploration Agency (JAXA) to generate yearly mosaics of the radar backscatter for each year between 2007 and 2010 (Shimada, 2010). Each image was orthorectified and radiometrically terrain corrected to gamma0 (Shimada, 2010). The mosaics are publicly available and are provided in the form of image tiles of 1° × 1° in latitude and longitude resampled to a grid with a pixel spacing of 0.000225°. In this study, we used the mosaics of the co-polarized

Horizontal-transmit Horizontal-receive (HH) and cross-polarized Horizontal-transmit Vertical-receive (HV) backscatter images. Images from 14,728 tiles were used to estimate biomass. 96 image tiles showing evident radiometric offsets with respect to adjacent ones, due for example to different environmental conditions (e.g., frozen vs. unfrozen conditions), were manually replaced with the corresponding image tile from the mosaic of 2009. This replacement ensured homogeneity of the L-band backscatter dataset across all landscapes.

Envisat ASAR was an active microwave sensor acquiring images at C-band (wavelength of 6 cm) between 2002 and 2012. ASAR operated in four different modes over land, with a spatial resolution of 30 m (Image Mode and Alternating Polarization Mode), 150 m (Wide Swath Mode) and 1,000 m (Global Monitoring Mode). Approximately 80% of the total number of observations consisted of GM observations. We processed the entire dataset of ASAR images of the SAR backscatter made available by the European Space Agency (ESA) through the Grid Processing on Demand platform to stacks of terrain geocoded, pixel-area normalized and speckle-filtered images (Santoro et al., 2015b). Images acquired with the IM and WSM were geocoded to a pixel size of 0.0013888° in latitude and longitude, corresponding to an area on the ground of roughly 150 m × 150 m at the Equator. Images acquired with the GMM were geocoded to a pixel size of 0.01° in latitude and longitude, corresponding to an area on the ground of approximately 1,000 m × 1,000 m at the Equator. To obtain global full coverage, the IM and WSM images were further averaged and resampled to the pixel size of the GMM dataset to form a single 1 km dataset of C-band backscatter observations. Each image was divided into tiles of 2° × 2° in latitude and longitude. For this study, we used all Envisat ASAR images acquired in 2010 and 2011. The density of the observations in time (Fig. S1) shows decreased from the polar latitudes with several observations per day to the tropical latitudes with approximately 200 observations on average over two years. Areas with a small number of observations correspond to regions seldom imaged during the lifetime of the Envisat mission (e.g. New Zealand, Japan) or imaged only at high resolution when the overlap of images from adjacent orbitals track was null (e.g., California, western Amazon).

The parameterization of the biomass retrieval models relating biomass to SAR backscatter observations was supported by the GLA14 data product of the Geoscience Laser Altimeter System (GLAS) on board the Ice Cloud and Land Elevation Satellite (ICESat) that operated between 2003 and 2009. GLA14 represented the waveforms over land only in the form of the parameters of a multi-Gaussian model fitted to the raw waveforms (Hofton et al., 2000), thus containing information about the vertical structure of vegetation. Because GLAS observations consisted of approximately 65 m large footprints acquired every 170 m along track with a distance between tracks of the order of 60 km, the GLA14 dataset was not dense enough to allow direct spatially explicit estimates of biomass. Here, we used the entire archive of GLA14 data products, provided by the National Snow & Ice Data Centre (NSIDC), to estimate forest height after filtering for footprints affected by topography and various noise sources in the waveforms (Los et al., 2012; Simard et al., 2011). In addition, we computed an estimate of the canopy density for each footprint as the ratio of energy received from the canopy (i.e., returns from above the ground peak) to the total energy received. Our database of GLAS-based metrics consisted of 26.5 million footprints homogeneously distributed over all

180 vegetated surfaces. While the SAR observations were used as predictors in the retrieval model (Section 3), the LiDAR observations supported the estimation of parameters of the retrieval model that are time invariant such as forest transmissivity (Section 3). Herewith, the time difference between the SAR imagery (2010) and the LiDAR observations (2003-2009) did not impact the retrieval.

Global reflectances of Landsat 7 images (bands 3, 4, 5 and 7) acquired in 2010 were used to rescale biomass estimates from ASAR to the pixel size of the ALOS PALSAR dataset. The dataset was available in the form of a mosaic from the USGS website (Hansen et al., 2013). The square pixels of the mosaic had a spacing of 0.00027°, i.e., roughly 30 m at the Equator. The dataset was downloaded from Google Earth Engine and resampled with nearest neighbour to the geometry of the ALOS PALSAR dataset.

**2.2 AGB estimation**

Unlike investigations that directly relate AGB to the remote sensing data (Lu et al., 2016; Santoro and Cartus, 2018), we estimated the density of the woody volume, referred to as growing stock volume (GSV, unit: $m^3$ $ha^{-1}$) from which AGB is then computed, for three reasons. First, the signal backscattered by a forest is primarily affected by the density, and to some degree the height, of the trees (Santoro et al., 2015a). However, the short wavelength of the radar instruments means that only the 195 upper part of the volume is seen so that an estimate of GSV (or AGB) would be the result of an inference from the SAR observation. Second, it has not been demonstrated that the SAR backscatter at C- and L-band is sensitive to the wood density of trees for a given level of GSV, thus not providing experimental support to a direct estimation of AGB. Without such evidence, it is preferable to estimate a forest structural parameter from the SAR backscatter and convert it to AGB using a separate layer combining the wood density and the stem-to-total biomass expansion factor, which does not depend on remote 200 sensing observations. An open question is whether means exist that allow an unbiased characterization of wood density and stem-to-total biomass expansion globally. Third, our approach to estimate AGB from remote sensing data mimics the approach based on forest field inventory data (Brown, 1987; Jenkins et al., 2003) where GSV is acknowledged to be the major predictor of AGB. The relevance of GSV to estimation of AGB is further emphasised by the country reports building up the FAO 2010 FRA (FAO, 2010). Of the 233 country reports, we identified 171 countries reporting numbers on AGB, and for roughly two 205 thirds of these (111) the estimate of AGB was derived from an estimate of GSV, based on inventory or expert knowledge, using a scaling factor.

The AGB retrieval algorithm is outlined in the flowchart in Fig. 1 showing the interdependencies of datasets and retrieval models. We applied a model-based approach known as BIOMASAR (Santoro et al., 2011; Cartus et al., 2012b) separately to 210 the ALOS PALSAR (Cartus et al., 2012b) and the Envisat ASAR radar backscatter datasets (Santoro et al., 2011, 2015a) to obtain two independent, spatially explicit estimates of GSV. BIOMASAR inverts a Water Cloud Model (Pulliainen et al., 1994; Santoro et al., 2002).

$$\sigma_{for}^0 = \sigma_{gr}^0 e^{-\beta V} + \sigma_{veg}^0 \left(1 - e^{-\beta V}\right) \tag{1}$$


$\sigma_{for}^0$ represents the forest backscatter, i.e., the SAR backscatter observation from an ALOS PALSAR or an Envisat ASAR image (Section 2.1). $\sigma_{gr}^0$ and $\sigma_{veg}^0$ represent the backscattering coefficients of the ground and vegetation layer, respectively. The exponential function, $e^{-\beta V}$, represents the two-way forest transmissivity, where $\beta$ is an empirically defined coefficient expressed in m$^{-1}$ and $V$ represents GSV. Eq. (1) neglects multiple scattering, which is acceptable for most forest conditions

(Santoro et al., 2011; Cartus et al., 2012a; Cartus and Santoro, 2019).

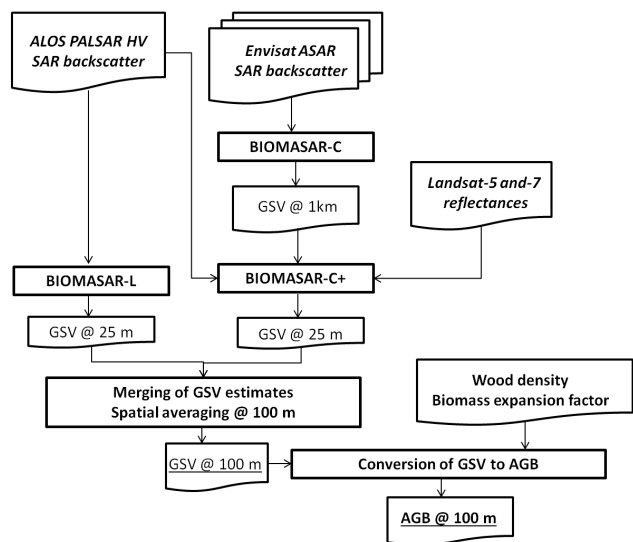

**Figure 1. Flowchart of the AGB retrieval approach.**

The model parameters $\beta$, $\sigma_{gr}^0$ and $\sigma_{veg}^0$ need to be estimated in order to invert the model and obtain an estimate of GSV from a measurement of the SAR backscatter. Estimates of the coefficient $\beta$ are obtained with a model-based approach that relates canopy density and GSV observations through the transmissivity of the forest (Santoro et al., 2015a). The estimation is stratified by the FAO Global Ecological Zones (Supplement Section A.1, Fig. S2 and Table S2). To estimate $\sigma_{gr}^0$ and $\sigma_{veg}^0$, we rely on a self-calibration approach (Santoro et al., 2011) rather than using a set of reference measurements of the SAR

backscatter and *in situ* GSV values. The limited availability of *in situ* information on biomass (e.g., from inventory plots or laser-based maps) prevents adaptive calibration of retrieval algorithms using conventional approaches. In many areas, particularly the tropics, the number of available plots is very limited so that models can only be calibrated using reference information collected over large areas (Bouvet et al., 2018) with the risk of missing spatial variability in the backscatter. The

model training approach is tailored to the radar wavelength in order to accommodate the different relationships of backscatter

to biomass (BIOMASAR-C and BIOMASAR-L, see Supplement Section A.2 and A.3).

When N observations of the radar backscatter are available, a final estimate of GSV, with higher accuracy compared to the individual estimates, is obtained by means of a weighted linear combination of the individual estimates of GSV obtained by inverting Eq. (1) for each backscatter observation (Santoro et al., 2011; Kurvonen et al., 1999).


$$V_{mt} = \frac{\sum_{i=1}^{N} w_i \hat{V}_i}{\sum_{i=1}^{N} w_i} \tag{2}$$

In Eq. (2), the weights $w_i$ are defined as the difference ($\sigma^0_{veg,i}$ - $\sigma^0_{gr,i}$) so that GSV estimates obtained from images with the strongest sensitivity to GSV are preferred to those obtained from images with no sensitivity to GSV (Santoro et al., 2011).


In this study, the BIOMASAR-C implementation was tailored to ingest ASAR data (Supplement Section A.2) and generated a GSV map at 1,000 m spatial resolution by combining individual estimates from the ASAR dataset. The BIOMASAR-L implementation was tailored to ingest ALOS PALSAR data and generate a GSV data product at 25 m spatial resolution (Supplement Section A.3). The step in Eq. (2) became redundant because of the strong correlation of the ALOS PALSAR

mosaics in time and the negligible weight attributed to the HH-polarized component when combined with the HV-polarized component (DUE GlobBiomass - Algorithm Theoretical Basis Document). As a result, the retrieved GSV with BIOMASAR-L was based on the single observation of the L-band SAR backscatter at HV-polarization in the mosaic for 2010.

To merge estimates of GSV obtained at 25 m and 1,000 m, the latter were re-scaled to 25 m using a linear regression model

(BIOMASAR-C+ in Fig. 1). The HH- and HV-polarized ALOS PALSAR backscatter ($\sigma^0_{HH}$ and $\sigma^0_{HV}$) and the Landsat bands 3,4,5 and 7 ($B_3$, $B_4$, $B_5$ and $B_7$) were used as predictors in the model in Eq. (3):

$$\log(V) = a_0 + a_1 \sigma^0_{HV} + a_2 \sigma^0_{HH} + a_3 B_3 + a_4 B_4 + a_5 B_5 + a_6 B_7 \tag{3}$$

The model was calibrated for each 1° × 1° tile at a pixel size of 1,000 m to predict the BIOMASAR-C estimate of GSV at the 25 m scale. A bias correction had to be performed when retransforming the logarithmic GSV predictions to linear scale. The bias was computed by differencing the original BIOMASAR-C GSV and the predictions from Eq. (3) aggregated to the 1,000 m pixel size. In spite of its simplicity, the multiple linear regression resulted in an overall superior performance when compared to results obtained with more sophisticated re-scaling methods (DUE GlobBiomass - Algorithm Theoretical Basis Document).


The final estimate of GSV was obtained by weighting the rescaled ASAR-based estimates ($V_{C+}$) and PALSAR-based estimates ($V_L$) of GSV. The weighting scheme accounted for the different sensitivity of C- and L-band data to GSV, the number of observations used for estimating GSV, local errors in the model training and inversion and residual uncompensated topographic effects in the ALOS PALSAR mosaics (Supplement Section A.4). The GSV estimated with BIOMASAR-L was given more weight in areas of high GSV except in the case of steep terrain (Fig. S3). BIOMASAR-C+ GSV estimates were instead preferred in areas of low GSV and regions with rugged terrain (Fig. S3).

$$GSV = V_{C+} \cdot w_C + V_L \cdot w_L \tag{4}$$

To reduce the pixel-wise variability due to speckle in the radar data and amplified by the weak sensitivity of the C- and L-band backscatter to forest variables, spatial averaging using a 4 x 4 window was applied. This decreased the spatial resolution of the GSV estimates to 0.000888° in both latitude and longitude, corresponding to an area of approximately 1 ha at the Equator.

The conversion of GSV to AGB in Eq. (5) requires wood density (WD) and the biomass expansion factors (BEF), which give the allometric relationship between stem mass and whole aboveground mass, including branches and leaves.

$$AGB = GSV \cdot WD \cdot BEF \tag{5}$$

The spatial variations in WD and BEF result from biological processes that respond to local conditions, as has been demonstrated in regional studies showing environmental controls on the patterns of WD and BEF (Chave et al., 2009; Thurner et al., 2014). Towards a global assessment of WD patterns, we collected published databases based on inventory data where wood density (WD) is reported and explored machine learning methods to maximize the information content in relevant environmental variables (Supplement Section A.5). The final dataset of WD was obtained by integrating several individual predictions (Supplement Section A.5). To estimate the BEF, we used the generalized power-law function relating branch and leaf biomass to stem biomass (SB) (Thurner et al., 2014).

$$BEF = \frac{p_1 SB^{p_2} + SB}{SB} \tag{6}$$

Because of the uneven distribution of samples for which biomass component measurements were available (Supplement Section A.5), the model in Eq. (6) was fitted to measurements of BEF and stem biomass for tropical and extra-tropical forests only, each stratified by leaf type (broadleaves, evergreen conifers and deciduous conifers). All BEF models decreased rapidly for increasing stem biomass reaching an asymptote for low to medium stem biomass depending on ecoregion and leaf type

(DUE GlobBiomass - Algorithm Theoretical Basis Document). The asymptotic BEF for tropical broadleaves species was modelled with a value of 1.36, higher than other forest types, which were characterized by values of 1.15-1.20.


Spatially explicit estimates of WD and BEF were obtained at 0.01° spatial resolution and resampled to the pixel size of the GSV dataset using bi-cubic interpolation. The AGB estimates resulting from the product of the GSV, WD and BEF estimates were obtained at a spatial resolution of 0.000888° in both latitude and longitude, i.e., with a pixel size of 1 ha at the Equator.

## 2.3 Uncertainty model

The uncertainty of the AGB estimates was quantified by their standard deviation. The standard deviation of the GSV estimates obtained with the BIOMASAR-C approach, $\delta V_C$, was quantified by propagating the standard deviation of the measured SAR backscatter, $\sigma^0_{meas}$, and the estimates of the forest backscatter model parameters $\sigma^0_{gr}$, $\sigma^0_{df}$, $\beta$ and $V_{df}$ (Santoro et al., 2015a).

$$\delta V_C = \sqrt{\begin{array}{c} (\delta\sigma^0_{meas})^2 \cdot \left(\frac{\partial V}{\partial\sigma^0_{meas}}\right)^2 + (\delta\sigma^0_{gr})^2 \cdot \left(\frac{\partial V}{\partial\sigma^0_{gr}}\right)^2 + (\delta\sigma^0_{df})^2 \cdot \left(\frac{\partial V}{\partial\sigma^0_{df}}\right)^2 + \\ \\ +(\delta\beta)^2 \cdot \left(\frac{\partial V}{\partial\beta}\right)^2 + (\delta V_{df})^2 \cdot \left(\frac{\partial V}{\partial V_{df}}\right)^2 \end{array}} \qquad (7)$$


The same approach was applied to the BIOMASAR-L procedure, in which case the error model also included components related to the average canopy density of dense forests, $\eta_{df}$, the average height of dense forests, $h_{df}$, and the two-way attenuation coefficient, $\alpha$. Following the results in (Simard et al., 2011; Los et al., 2012), which validated GLAS-based height estimates at boreal, temperate, sub-tropical, and tropical forest sites, we assumed standard deviations for height estimates at the GLAS

footprint-level between 4 m (boreal zone) and 10 m (tropical zone). As indicated by Garcia et al. (2012), the estimation error of canopy cover from ICESAT GLAS as the ratio of energy returned from the canopy to the total energy returned may be of the order of 15 to 20 %. We therefore assume a global error of 20%. Differently than for the C-band case (Santoro et al., 2015a), the standard deviation of the coefficient $\beta$ was inferred from the relationship of the forest transmissivity, simulated with the aid of GLAS height and optical canopy density estimates, and GSV (DUE GlobBiomass - Algorithm Theoretical

Basis Document). The 95% bounds of the estimates increased from +/-0.002 ha/m$^3$ in the case of low values of $\beta$ that are valid in boreal and subtropical dry forests to +/-0.007 ha/m$^3$ for the highest values of $\beta$ that are applied in the tropics. For the two-way attenuation coefficient $\alpha$, we assume a standard deviation of 0.25 dB/m, which is roughly consistent with the range of values reported in the literature (DUE GlobBiomass - Algorithm Theoretical Basis Document).

$$\delta V_L = \sqrt{\begin{array}{l} (\delta\sigma_{meas}^0)^2 \cdot \left(\frac{\partial V}{\partial \sigma_{meas}^0}\right)^2 + (\delta\sigma_{gr}^0)^2 \cdot \left(\frac{\partial V}{\partial \sigma_{gr}^0}\right)^2 + (\delta\sigma_{df}^0)^2 \cdot \left(\frac{\partial V}{\partial \sigma_{df}^0}\right)^2 + \\[2mm] + (\delta h_{df})^2 \cdot \left(\frac{\partial V}{\partial h_{df}}\right)^2 + (\delta\eta_{df})^2 \cdot \left(\frac{\partial V}{\partial \eta_{df}}\right)^2 + (\delta\alpha)^2 \cdot \left(\frac{\partial V}{\partial \alpha}\right)^2 \end{array}} \tag{8}$$

In the case of BIOMASAR-C, the standard deviation of the multi-temporal GSV estimate was modelled as a linear combination of the single-image GSV standard deviations from Eq. (9) (Santoro et al., 2015a).

$$\delta V_{C,mt} = \sqrt{\sum_{i=1}^N w_i^2 \, \delta(V_{C,i})^2} \tag{9}$$

The uncertainty associated with the predictions of GSV obtained by rescaling the BIOMASAR-C GSV estimates was related to the uncertainty in the coarse resolution GSV estimates and the scaling factors.

The relative standard deviation of the BIOMASAR-C GSV estimates ($\delta V_C/V_c$) was modelled as a function of the GSV estimates ($V_C$) by means of an exponential model (DUE GlobBiomass - Algorithm Theoretical Basis Document). In Eq. (10), the model coefficients $a$, $b$ and $c$ were estimated by means of a least squares regression for each of the FAO Global Ecological Zones.

$$\delta V_C/V_C = ae^{bV_C} + c \tag{10}$$


To characterize the error associated with the rescaling model, we used the root mean square difference (RMSD) between the original BIOMASAR-C GSV and the BIOMASAR-C+ GSV estimates aggregated to the pixel size of the former. A polynomial function of the fourth order was found to adequately reproduce the relationship between the GSV sets across ecozones and was used to fit the observed trend in GSV (DUE GlobBiomass - Algorithm Theoretical Basis Document).


In the process of rescaling, we assumed that the standard deviation scaled with the pixel area of the GSV predictions. In Eq. (11), the scaling factor between standard deviations was represented by the ratio between the pixel areas of the rescaled product ($A_{c+}$) and the BIOMASAR-C data product ($A_c$).

$$\delta V_{C+} = \delta V_C \sqrt{\frac{A_{C+}}{A_c}} \tag{11}$$

The standard deviation of the GSV estimates was obtained with the same weighted linear combination of the standard deviations of the BIOMASAR-L and the BIOMASAR-C+ datasets:

$$\delta GSV = \delta V_{C+} \cdot w_C + \delta V_L \cdot w_L \tag{12}$$

The standard deviation of AGB was expressed in terms of partial derivatives of its components

$$\delta AGB = \sqrt{\left(\frac{\partial AGB}{\partial WD}\right)^2 \cdot \delta WD^2 + \left(\frac{\partial AGB}{\partial BEF}\right)^2 \cdot \delta BEF^2 + \left(\frac{\partial AGB}{\partial GSV}\right)^2 \cdot \delta GSV^2} \tag{13}$$


The standard deviation of the wood density estimates, $\delta WD$, was obtained by computing the variance of the predictions for each measurement of wood density and fitting a linear model (Fig. S4). The standard deviation of the BEF was expressed in terms of partial derivatives of its components

$$\delta BEF = \sqrt{\left(\frac{\partial BEF}{\partial p_1}\right)^2 \cdot \delta p_1{}^2 + \left(\frac{\partial BEF}{\partial p_2}\right)^2 \cdot \delta p_2{}^2 + \left(\frac{\partial BEF}{\partial SB}\right)^2 \cdot \delta SB^2} \tag{14}$$

In Eq. (14), $\delta p_1$ $and$ $\delta p_2$ represent the standard deviations of the two coefficients of the BEF model.

**2.4 Validation**

To assess the accuracy of the AGB estimates, we used a reference dataset of AGB observations from 110,897 forest field
inventory plots. The data were gathered from a variety of surveys undertaken by national forest inventories and research networks (Supplement Section A.6).

The opportunistic nature of our validation database led to an uneven spatial distribution of the reference samples (Fig. S5), as well as a variety of plot sizes, survey methods and allometric equations used (Table S3 and Table S4). The plots were mostly
smaller than 1 ha (Table S3), implying that they often represented a small fraction of the area covered by a 1-ha map pixel. To reduce the effect of random errors caused by the mismatch in resolution between the reference dataset and our map model, we aggregated our map and the plot data to 0.1° grid cells. This represented as a trade-off between capturing the local scale variability of AGB whilst allowing a number of plot measurements deemed sufficient to compute an average AGB representative of the area within the grid cell. In the end, our assessment cannot provide an indication of the validity of pixel-
based AGB estimates. Instead, it provides a measure of the accuracy of generalized spatial AGB patterns. This is, however, a pragmatic approach when using measurements not designed for validation of estimates from remote sensing imagery but which often provide the only source of information on AGB in poorly inventoried regions or regions where national inventory data are not publicly available.

To provide a more comprehensive overview of the reliability of the spatial patterns in areas not covered by the database of plot inventory measurements, the analysis was supplemented by a comparison of average GSV or AGB at the level of inventory reference units (polygons, counties, provinces and ecoregions). The scope of this analysis was primarily to identify systematic errors, on a large scale, that may not become evident when comparing at individual plot level. For a quantitative assessment of the retrieval at the scale of provincial and regional aggregates, we computed the RMSD between map and reference biomass averages relative to the average reference biomass and the bias between map and reference biomass averages. The RMSD was computed as a weighted mean of the errors, where the weights corresponded to the ratio of the forest area to the total forest area. For Russia (Supplement Section A.6), GSV and AGB data were gathered for approximately 1,600 Forest Managements Units (FMU) ranging in size from 3,000 ha (e.g. intensive forestry or national parks in the European part) to 30,000,000 ha (remote territories in Siberia). For countries with a well-established national forest inventory that regularly publish regional statistics of forest biomass at the level of administrative or ecological units, we assembled a database of GSV and AGB averages representative of the epoch 2010 (Supplement Section A.6).

## 2.5 Inter-comparison of AGB maps

The spatial distribution of AGB map estimates from our dataset were compared with biome and global forest AGB maps based on satellite remote sensing observations (Saatchi et al., 2011b; Baccini et al., 2012; Thurner et al., 2014; Liu et al., 2015; Avitabile et al., 2016; Santoro et al., 2015a) or ancillary datasets (Kindermann et al., 2008). While an additional assessment of our dataset against regional maps of AGB would further contribute to build confidence in the data product, it is felt that such investigations require an own framework also making use of local reference data and expert knowledge as for example undertaken for Europe (Avitabile and Camia, 2018), the United States (Spawn et al., 2020) and Tanzania (Næsset et al., 2020). To do this, the datasets were first harmonized to a common geographic map projection and resampled to a pixel size of 0.01°. Datasets expressing AGB in carbon units (MgC ha$^{-1}$) were converted to AGB using a carbon fraction default value of 0.47 (IPCC, 2006) (Table S5). All datasets span a decade of input observations from 2000 to 2010 but could not be harmonized to reflect the conditions of a single epoch due to the lack of information on growth rates. This, however, is not expected to have significant effects on the interpretation of the latitudinal profiles.

As a way of assessing the AGB patterns of each map, we compared latitudinal averages based on values from the 0.1° validation grid cells with corresponding values from the database of forest inventory plots. To obtain a homogeneous representation of all latitudes, we grouped grid cells in 10° wide intervals.

# 3. Results

## 3.1 Global AGB dataset

The global AGB dataset (Fig. 2) was obtained by scaling global estimates of GSV (Fig. S6) using model-based estimates of wood density (Fig. S7) and stem-to-total biomass expansion factors (Fig. S8). The uncertainty of the AGB estimates is reported as standard deviation (Fig. 2). The models for retrieving GSV and converting it to AGB were developed for woody vegetation, so we evaluated the estimates corresponding just to forest cover by regrouping the classes from the Climate Change Initiative Land Cover (CCI-LC) dataset of 2010 into forest and non-forest land (Supplement Section A.1, Table S7).

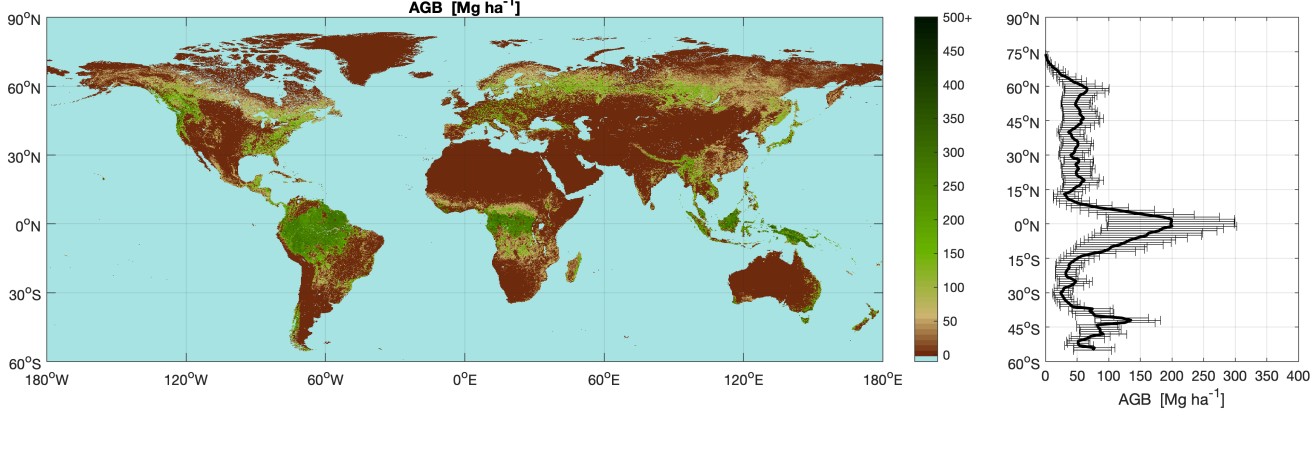

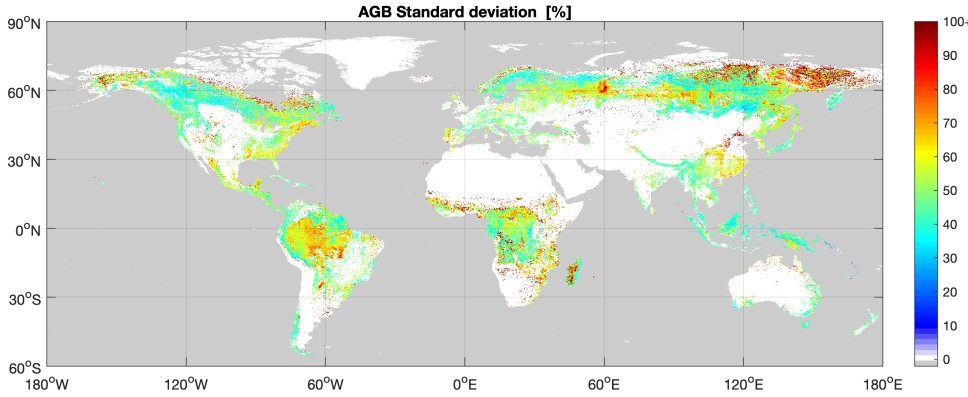

**Figure 2. Map estimates of AGB (top panel) and AGB standard deviation expressed relative to the AGB (bottom panel). The colour bar of the AGB map has been truncated at 500 Mg ha$^{-1}$ to increase contrast. Similarly, the colour bar of the AGB relative standard deviation has been truncated at 100%. The right hand panel shows the profile of average AGB along latitude (thick solid line) and the two-sided average standard deviation of AGB at a given latitude (horizontal bars).**

At the 1-ha scale, the largest predicted value was 757 Mg ha$^{-1}$, corresponding to a GSV of 1,087 m$^3$ ha$^{-1}$, in forests of the U.S. Pacific Northwest. However, for 99% of the world's forests, AGB was estimated to be less than 360 Mg ha$^{-1}$, and 90% was below 182 Mg ha$^{-1}$, which explains truncating the colour bar in Fig. 2 at 500 Mg ha$^{-1}$. The spatial distribution of AGB followed a clear latitudinal gradient (Fig. 2). In the northern hemisphere, AGB increased steadily with decreasing latitude across the boreal zone (between 75°N and 60°N), then remained fairly constant throughout the temperate (between 60°N and 40°N) and sub-tropical (between 40°N and 20°N) zones. AGB increased sharply as we enter and leave the tropical zone between 20°N and 20°S, though with a minimum at 13°N due to the large area of low biomass dry forests in the sub-Sahelian region. The AGB of semi-tropical forests in the southern hemisphere between 20°S and 33°S was slightly lower than in the corresponding latitude range in the northern hemisphere because of the larger proportion of low-density forest. The local maximum at 25°S corresponded to the Atlantic forests of Brazil and dense sub-tropical forests along the east coast of Australia where biomass accumulation is favoured by higher precipitation. Temperate forests had higher AGB in the southern hemisphere (south of 33°S) than in the northern hemisphere because of the predominant highly productive evergreen and coniferous forest along the Chilean-Argentinean Andes, in south-eastern Australia and New Zealand. The peak at 42°S corresponds to the broadleaved forests of Tasmania.

The AGB standard deviation, expressed in Fig. 2 relative to the AGB estimates, was on average 50% with an inter-quartile range of values between 44% und 61%. The rather constant relative uncertainty is also illustrated by the horizontal bars in the latitudinal profile of Fig. 2, which scale with the AGB level. The relative standard deviation was smaller than 100% for approximately 95% of the mapped pixels which explains truncating the colour bar of the AGB standard deviation map in Fig. 2 at 100%. The large majority of the AGB estimates for which the standard deviation exceeded the 100% level were below 20 Mg ha$^{-1}$ such as in sparsely vegetated regions corresponding to the transition to tundra in Canada and Alaska, the Siberian Lowlands and Far East Russia or in poorly stocked forests such as in northern China and west Madagascar.

The small-scale variability of AGB in forest landscapes is captured by the 1-ha pixel spacing of the dataset (Fig. 3 and Fig. 4). The example in Fig. 3 shows the region of the Bratsk Reservoir formed by the Angara River in Central Siberia, where forests are heavily managed for timber production. Clear-felling activities occur in polygons often larger than 10 ha. The map for a 1° × 1° area east of the reservoir shows forests with AGB above 100 Mg ha$^{-1}$, roughly corresponding to a GSV of at least 200 m$^3$ ha$^{-1}$, interspersed with small rectangular white shapes corresponding to clear-cut areas (Fig. 3). Our estimates give more detailed information on the spatial patterns of AGB in this region than the AGB product by Thurner et al. (Thurner et al., 2014) based on remote sensing data with a spatial resolution of 1,000 m.

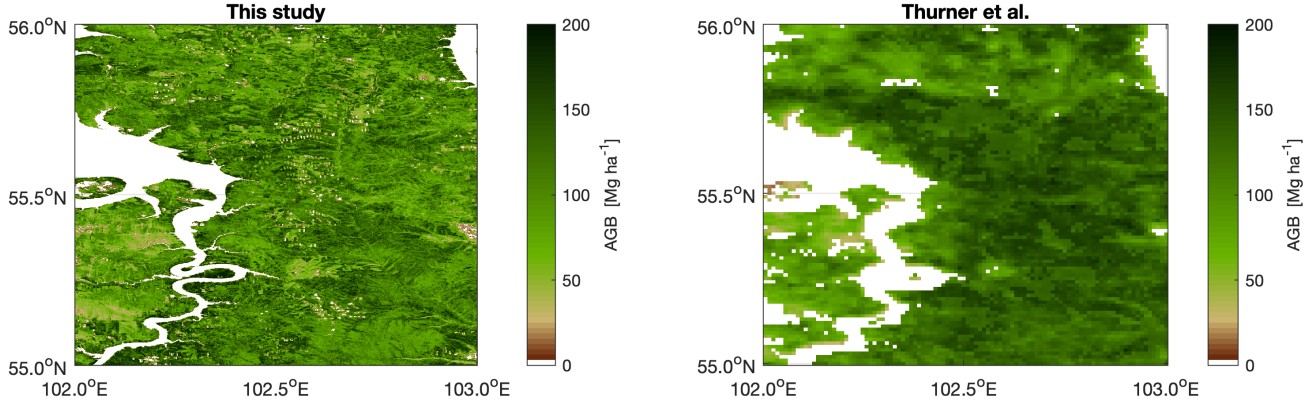

Figure 3. AGB estimates from this study (left) and from Thurner et al. (Thurner et al., 2014) for a 1° × 1° area in Central Siberia.

The example in Fig. 4 includes part of Trans-Amazonian Highway south of the Amazon River in the state of Pará, Brazil. This region is characterized by fishbone deforestation caused by lateral expansion of agriculture from the highway into pristine forest. The fishbone pattern is clear in our AGB map (Fig. 4, top left panel) and some isolated, small-scale patches of deforestation are also visible. For comparison, Fig. 4 shows AGB estimates by three pan-tropical maps based on remote sensing observations (Saatchi et al., 2011b; Baccini et al., 2012; Avitabile et al., 2016). The deforestation patterns are less extended in the map by Saatchi et al. (2011b) because it was based on satellite data acquired around the year 2000, compared to the maps by Baccini et al. (2012), based on observations taken in 2007, and Avitabile et al. (2016), which merged the other two pan-tropical datasets. The level of detail in our map is much greater because of the 1-ha spatial resolution compared to 25 ha in Baccini et al. and nearly 100 ha in Avitabile et al. and Saatchi et al.

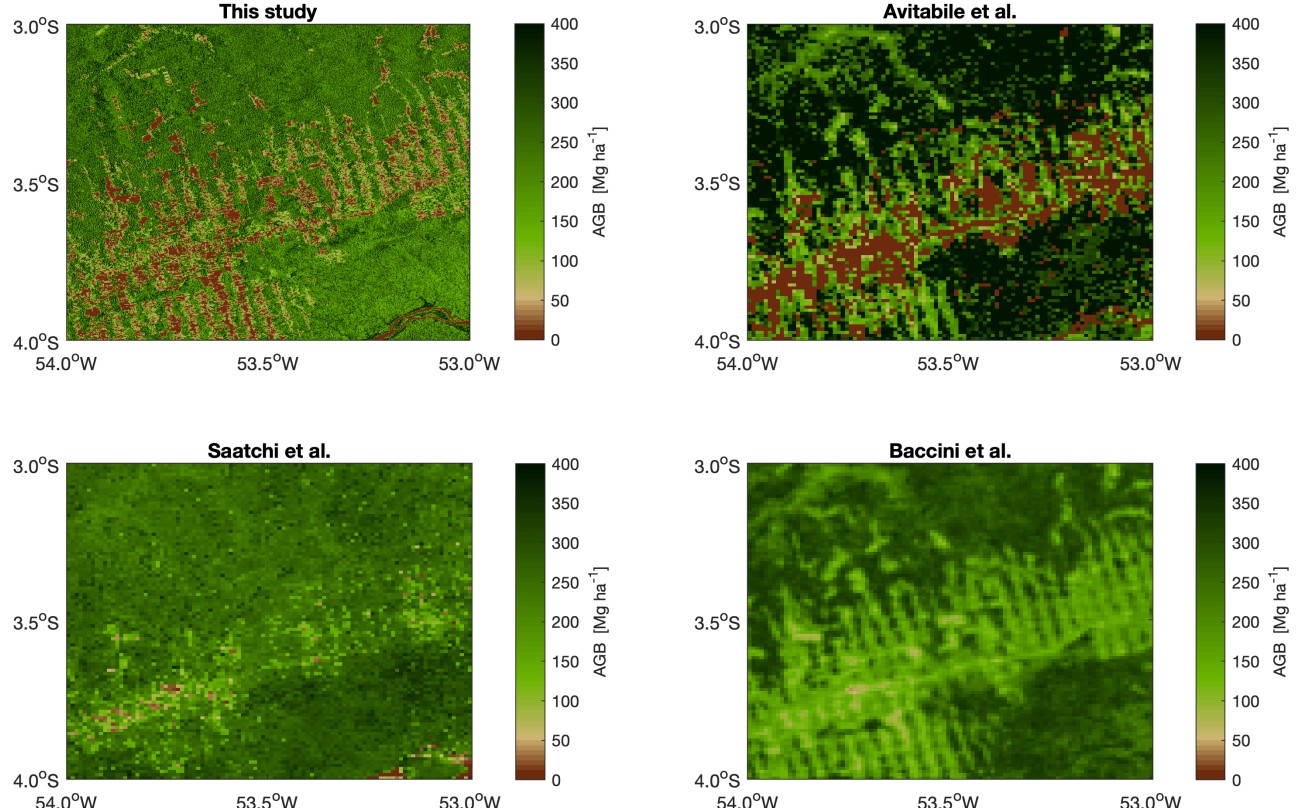

**Figure 4. AGB estimates by this study (top left), Avitabile et al. (2016) (top right), Saatchi et al. (2011b) (bottom left) and Baccini et al. (2012) (bottom right) for a 1° × 1° area in the state of Parà, Brazil.**

## 3.2 Validity of AGB estimates

AGB averages were obtained for 6,456 0.1° grid cells including at least five inventory plots (Supplement Section A.6). These grid cells represented approximately 1% of the Earth's forest cover. The grid cell average AGB from the field inventory database ranged from 0 to 1,670 Mg ha$^{-1}$ (median: 43 Mg ha$^{-1}$, mean: 60 Mg ha$^{-1}$, 99$^{th}$ percentile: 351 Mg ha$^{-1}$). The AGB histogram was skewed towards low values (Fig. 5a) because of the large proportion of measurements from the National Forest Inventories of Spain and Sweden (Table S4). The grid cell average AGB from the map ranged from 0 to 358 Mg ha$^{-1}$ (median: 57 Mg ha$^{-1}$, mean: 67 Mg ha$^{-1}$, 99$^{th}$ percentile: 278 Mg ha$^{-1}$). The AGB histograms from the map (Fig. 5b) and the field inventory (Fig. 5a) were similar, with a mode at around 0, a decline to a shoulder, then a further decline to a long tail. In the field inventory the shoulder covered the range 25-50 Mg ha$^{-1}$ and was followed by a slow decline, but for the map it extended to around 80 Mg ha$^{-1}$ and then declined rapidly. This difference arises because the map tends to give higher values than inventory in the lower AGB range (Fig. 5c and Fig. S9). These trends have been reported for other pan-tropical and regional AGB studies (Avitabile et al., 2016; Rodríguez-Veiga et al., 2019). The scatterplot of map against inventory values of AGB in Fig. 5c and the RMSD curve in Fig. S9 indicate an agreement in trend between field inventory and map values up to about

250 Mg ha[-1]. Above 250 Mg ha[-1], the map values rose as the field inventory value did, albeit more gently and with much greater variance. Disaggregating the data by major ecological domains, using the FAO Global Ecological Zones as reference, suggested slight differences in the agreement between map and inventory values in tropical, sub-tropical and temperate forests (Fig.s 5d-f).

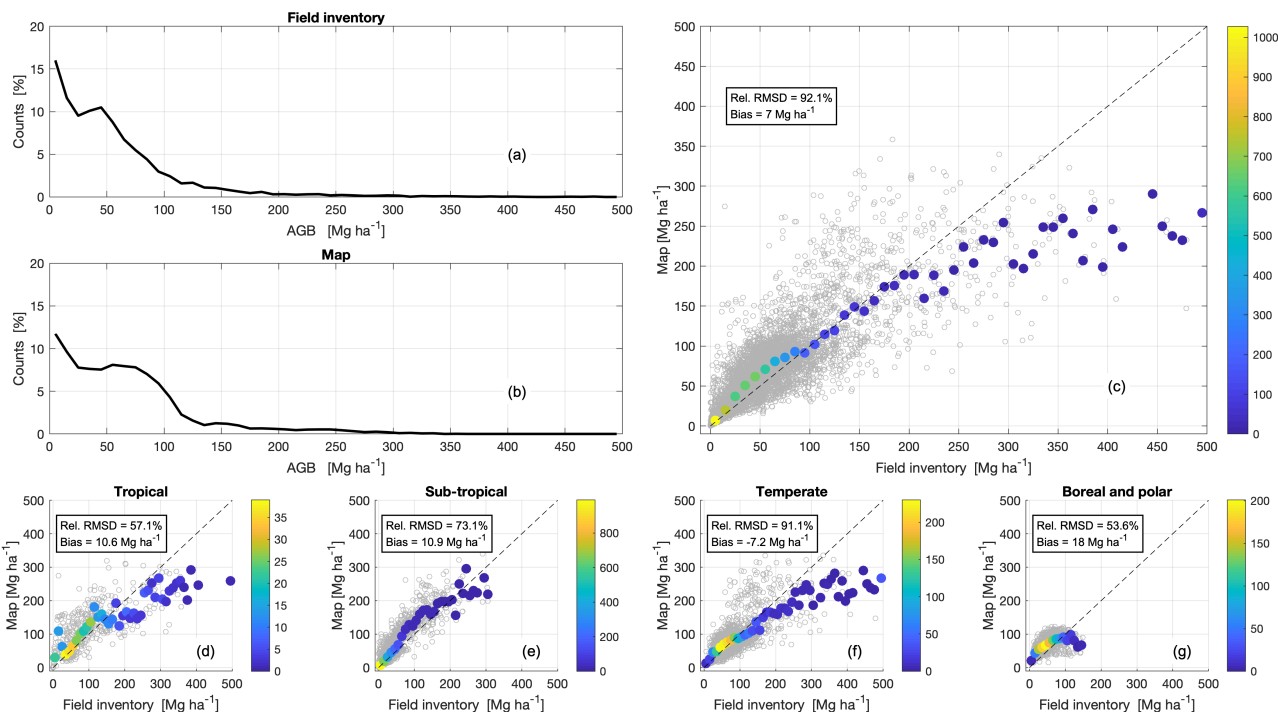

**Figure 5.** Histograms of AGB from the field inventory database (a) and the map (b) for 0.1° grid cell values. (c) Scatterplot of map AGB against field inventory values for 0.1° grid cells (grey circles); the filled circles show the median AGB of the map values in each 10 Mg ha[-1] wide interval of field inventory AGB values. The colour bar represents the number of grid cells within a given AGB interval. Similar scatterplots are given for the tropical zone (d), the sub-tropical zone (e), the temperate zone (f) and the boreal and polar zones (g) according to the FAO Global Ecological Zones. On each scatterplot, we report the root mean squared difference (RMSD) between map and field inventory AGB relative to the mean value of the reference AGB and the bias, i.e., the difference between mean values of the map AGB and the reference AGB. To improve presentation and because of the paucity of grid cells with AGB above 500 Mg ha[-1], axes are truncated at 500 Mg ha[-1].

## 3.3 Spatial distribution of AGB

Combining our AGB dataset with the CCI Land Cover dataset and the FAO ecological zones, we estimated a total AGB of 521 Pg for a forest area of 4,825 Million ha, corresponding to a global average forest AGB of 108 Mg ha[-1] (Table 1). Using the carbon fraction default value of 0.47 (IPCC, 2006), we estimated a total above-ground carbon stock of 246 PgC, Tropical forests had the highest average AGB (147 Mg ha[-1]), representing 64% of the total forest AGB and 47% of the total forest area. The second largest average AGB was found in temperate forests (102 Mg ha[-1]), which accounted for 14% of the total AGB and 15% of the total forest area. Sub-tropical and boreal forests had similar average AGB (75 and 60 Mg ha[-1], respectively),

but the area covered by the latter was almost three times larger. As a result, the total AGB of boreal forests was more than twice as large as that of sub-tropical forests and corresponded to 16% of the total AGB, thus being larger than the AGB pool in temperate forests. The total AGB of subtropical forests accounted for 7% of the global AGB. The contribution of polar forests to the global AGB pool was negligible (0.1%).

Looking at Fig. 6, the tropical rainforest (TAr) ecozone hosted primarily high-density forests, with a median AGB of 238 Mg ha$^{-1}$ and the low-end of the interquartile range above 150 Mg ha$^{-1}$. Besides tropical rainforests, only the tropical mountain, temperate oceanic and temperate mountain ecozones (TM, TeDo and TeM, respectively) had median AGB value above 100 Mg ha$^{-1}$ (Fig. 6). For the tropical, sub-tropical and temperate ecozones, the AGB of forests in dry environments (shrubland, steppe and desert) was lower than in wet environments (rainforest, moist, humid, mountain, continental and oceanic). The AGB of boreal forests decreased with increasing latitude from the boreal coniferous (Ba) ecozone located at the southernmost edge of the boreal ecotone, through the boreal mountain (BM) ecozone, to the boreal tundra woodland (Bb) ecozone at the northernmost edge of the boreal zone.

The AGB standard deviation relative to the AGB estimates was fairly constant across most ecological zones (Fig. 6). The median value ranged between 44% and 57%, except for the tropical shrubland, tropical desert and polar ecozones (TBSh = 69%; TBWh = 92% and P = 84 %, respectively). The largest proportion of the AGB standard deviation was attributed to the uncertainty of the GSV estimates (Fig. S10). The uncertainty of the wood density estimates accounted for 7% to 20% (mean: 14%) of the AGB standard deviation (Fig. S10), while the uncertainty in the BEF accounted for between 2% to 13% of the AGB standard deviation (mean: 7%) with the exception of the tropical desert zone (30%) (Fig. S10). The uncertainty of the GSV was driven by the weak sensitivity of the radar backscatter to increasing GSV, an effect further exacerbated in wet environments (Santoro et al., 2015a), thus explaining the slightly higher uncertainty in the tropics and the sub-tropics. The larger uncertainty in sparsely forested regions compared to densely forested regions (Fig. 2) is a consequence of the substantially larger uncertainty of the $\sigma^0_{gr}$ parameter compared to the uncertainty of the $\sigma^0_{veg}$ in the GSV retrieval model in Eq. (1) (Fig. S11). In sparse forests or forests with low woody biomass stocks, the total backscatter from the forest is dominated by the ground scattering component, i.e., the term with $\sigma^0_{gr}$ in Eq. (1), thus being affected by larger uncertainty compared to the backscatter received from dense forests, for which the largest contribution to the measured backscatter is attributed to the scattering from the canopy, i.e. to the term with $\sigma^0_{veg}$ in Eq. (1).

**Table 1. Total AGB, forest area and average AGB per major ecozone.**

| Ecozone | Total AGB (Pg) | Forest area ($10^6$ ha) | Average AGB (Mg ha$^{-1}$) |
|---|---|---|---|
| Tropical | 331.3 | 2251.7 | 147 |
| Sub-tropical | 36.2 | 483.0 | 75 |
| Temperate | 71.6 | 698.3 | 102 |
| Boreal | 81.2 | 1352.5 | 60 |
| Polar | 0.6 | 39.8 | 18 |
| Total | 521.0 | 4825.4 | 108 |

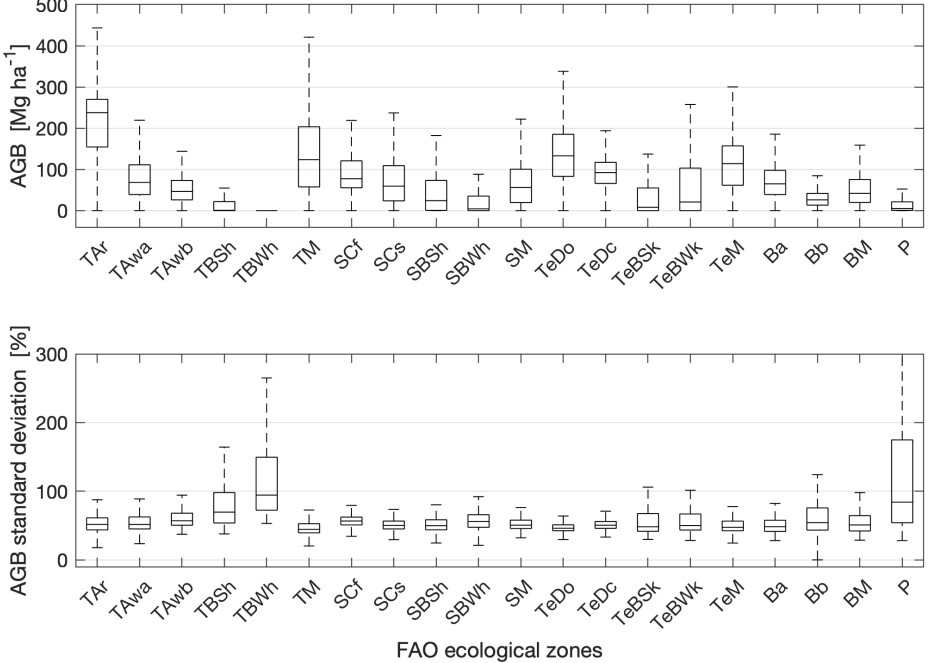

**Figure 6. Box plot diagram of AGB (top) and its standard deviation by FAO global ecological zone. On each box, the central mark**
**indicates the median, the bottom and top edges delimit the interquartile range, and the whisker delimits the 1-99 percentile range. Labels of global ecological zones: TAr: Tropical rain forest; TAwa: Tropical moist deciduous forest; TAwb: Tropical dry forest; TBSh: Tropical shrubland; TBWh: Tropical desert; TM: Tropical mountain systems; SCf: Subtropical humid forest; SCs: Subtropical dry forest; SBSh: Subtropical steppe; SBWh: Subtropical desert; SM: Subtropical mountain systems; TeDo: Temperate oceanic forest; TeDc: Temperate continental forest; TeBSk: Temperate steppe; TeBWk: Temperate desert; TeM: Temperate**
**mountain systems; Ba: Boreal coniferous forest; Bb: Boreal tundra woodland; BM: Boreal mountain systems; P: Polar.**

## 3.4 Assessment of global forest biomass resources

The most comprehensive summary of global forest resources and biomass pools is reported by the FAO in their quinquennial FRA. For the FRA, each country reports its values of forest area (in hectares), total AGB (in Pg) and average AGB (in Mg ha$^{-1}$) according to their inventory capabilities. Forest area was derived from inventory data or remote sensing data. AGB statistics were derived either from measurements collected as part of national inventories, local inventories or estimates reported in the literature. Adjustments were applied by the FAO where necessary to ensure consistency with its own information sources on forest area and biomass resources. While all 233 countries contributing to the FRA for 2010 reported their estimates of forest area, only 171 reported total AGB and average AGB. Of the remaining 62 countries, accounting for approximately 2.6% of the global land surface, 10 reported no forest cover, while 7 (Fiji, Eritrea, Uruguay, Ecuador, Paraguay, Japan and Venezuela) reported an estimate of forest area exceeding $10^6$ ha. Of the countries reporting on AGB, 111 derived their average AGB from values of the average GSV using one or multiple BCEF values. This was common practice for countries lacking a systematic national forest inventory. Among these, 79 relied on default numbers published by the Intergovernmental Panel on Climate Change (IPCC) (IPCC, 2006) (Table S7).

### 3.4.1 Global and continental statistics

The statistics on forest area, average AGB and total AGB from the FRA 2010 and from the combination of our AGB dataset with the CCI Land Cover dataset are reported by continent (Table 2). Our total AGB estimate of 522 Pg was 11% higher than the value of 468 Pg reported by the FRA. This difference is a consequence of the 23% larger forest area estimated from the CCI Land Cover dataset (Table 2) despite the FRA reporting 9% higher global average AGB than our estimate (119 Mg ha$^{-1}$ vs. 108 Mg ha$^{-1}$, Table 2). Compared to Table 1 based on the FAO Global Ecological Zones dataset to delineate land surfaces, we estimate an additional 8 $10^6$ ha of forest area and an additional 1.5 Pg of total AGB as a consequence of the more precise delineation of the land surface by the Database of Global Administrative Areas used as reference for the countries boundaries (GADM, http://www.gadm.org).

For the three continents spanning the tropics, we found the highest average AGB in South America, although our estimate (183 Mg ha$^{-1}$) was 11% lower than the corresponding value from the FRA (206 Mg ha$^{-1}$) (Table 2). South America also contained the largest total AGB pool. Although our estimate (155.9 Pg) was only 4% smaller than the FRA (162.3 Pg), the FRA did not provide AGB for approximately 10% of the forest area of South America (Table 2). We found a larger difference between our results and those of FAO for Africa. The average AGB from our dataset was 24% lower than the FRA (108 Mg ha$^{-1}$ vs. 142 Mg ha$^{-1}$, Table 2) whilst the total AGB was only 11% smaller than in the FRA (84.8 Pg vs. 95.3 Pg) due to the larger forest area we used from the CCI Land Cover dataset. In contrast, for Asia the average AGB from our dataset (115 Mg

ha$^{-1}$) was 17% higher than in the FRA (98 Mg ha$^{-1}$) (Table 2) while our estimate of total AGB exceeded that from the FRA by 64% (89.4 Pg vs. 54.3 Pg) mainly because of the 40% larger forest area estimated from the CCI Land Cover dataset.

For the two continents spanning the northern boreal and temperate zones, the average AGBs from our estimates were well below 100 Mg ha$^{-1}$ (Table 2). The average AGB estimate for Europe differed by less than 1% (Table 2) compared to the FRA whereas for North/Central America the difference was 15%. Because the forest area estimated from the CCI Land Cover dataset was larger than the FRA values, the total AGB estimated from our dataset was larger than the values reported in the FRA by 20% for Europe and 14% for North/Central America.

Finally, the smallest of the continental pools of AGB was in Oceania, where our larger estimate of total AGB compared to the FRA was primarily explained by our estimate of AGB being on average almost 35% larger than the value derived from the FRA.

**Table 2. Total AGB (unit: Pg), forest area (unit: ha) and average AGB (unit: Mg ha$^{-1}$) per continent from this study and from the FAO FRA 2010. The forest area column for the FRA does not account for countries reporting forest area but not AGB. For Asia, North/Central America and South America, 5%, 1% and 10% of the forest area did not contribute to the AGB to the FRA. For Africa, Europe and Russia, and Oceania, less than 1% of the forest area did not contribute to the FRA. The total forest area from the FRA is 4.033 10$^6$ ha.**

| Continent | This study | | | FAO FRA 2010 | | |
|---|---|---|---|---|---|---|
| | Total AGB (Pg) | Forest area (10$^6$ ha) | AGB (Mg ha$^{-1}$) | Total AGB (Pg) | Forest area (10$^6$ ha) | AGB (Mg ha$^{-1}$) |
| Africa | 84.8 | 783.5 | 108 | 95.3 | 672.6 | 142 |
| Asia | 89.4 | 780.5 | 115 | 54.3 | 554.0 | 98 |
| Europe and Russia | 91.2 | 1268.3 | 72 | 73.7 | 1016.5 | 72 |
| North and Central America | 77.9 | 970.6 | 80 | 66.8 | 704.4 | 95 |
| South America | 155.9 | 850.0 | 183 | 162.3 | 788.9 | 206 |
| Oceania | 23.3 | 180.4 | 129 | 16.0 | 189.7 | 85 |
| Total | 522.5 | 4,833.4 | 108 | 468.5 | 3,926.1 | 119 |

### 3.4.2 National statistics

At the level of individual countries, the agreement between the total and average AGBs from out dataset and from the FRA differed depending on the continent (Fig. 7 and Fig. 8, Table S7). The largest difference between our average AGB and FRA

AGB was in Africa (median difference: -60%, 51 countries), where for most countries the average AGB reported in the FRA exceeded the value from our dataset (Fig. 7). This is probably due to either AGB underestimation in countries dominated by high-density forests or to high biomass conversion and expansion factors (BCEF) used by countries with low-density forest when estimating values of AGB from their original measurements of GSV (Table S7). In addition, several countries used a small sample of plots for the calculations, as well as small-sized plots for heterogeneous forest areas, leading to large uncertainties in the values reported to the FRA. In Europe and South America, where the span of the average AGB was similar to Africa, we also saw underestimation patterns above 200 Mg ha$^{-1}$ (Fig. 7). Nonetheless, we did not identify the trend in the range 0 - 200 Mg ha$^{-1}$ seen in Africa. For European and South American countries, the assessment of the reliability of our AGB averages at country level was more meaningful because of the better developed national inventories (Fig. 8b) and the direct estimation of AGB from the inventory measurements, thus bypassing the use of a standard BCEF like in African countries (Table S7). Indeed, the smallest differences between the average AGB from the FRA and from our dataset were obtained in Europe (median difference: -8%, 42 countries) and South America (median difference: 2%, 9 countries) (Fig. 8a).

For North/Central America (19 countries), Asia (35 countries) and Oceania (9 countries), the median difference between our average AGB estimates and the FRA numbers was between -23% and -27% (Fig. 7, Table S7). The disagreement for North/Central America was largest for the Caribbean countries (Fig. 8a), for most of which the reference data used in the FRA had low to moderate quality and the NFI capacity was mostly low or limited.

The scatterplot for Asian countries shows data points of average AGB clustered along the identity line (Fig. 7) but with two distinct regions. For the Asian Middle East stretching as far as Pakistan and the former Soviet countries of the Asian continent, the average AGB from our dataset was on average 70% smaller than the values reported in the FRA (Fig. 8a). The FRA country reports were based on highly stocked forest, which may not be representative of the true average AGB. In contrast, the average AGB estimated from our dataset was approximately 27% larger than the values reported in the FRA for the southern and eastern regions of the Asian continent (Fig. 8a). Several countries of Southeast Asia assumed their forests to be strongly degraded which justified the use of low reference values for the average GSV, and hence AGB, when reporting to the FRA.

Our estimates of average AGB for Australia (brown marker, Fig. 7, panel Oceania) and Papua New Guinea (green marker, Fig. 7, panel Oceania) exceed the values in the FRA, while being smaller for New Zealand (cyan marker, Fig. 7, panel Oceania). Australia reported their biomass stock based on models calibrated with a small number of inventory measurements. For Papua New Guinea the FRA AGB was based on commercial volume for trees with a diameter at breast height of at least 50 cm, thus being a fraction of the true AGB. For New Zealand, the result was comparable to those obtained for European countries with large AGB, which are characterized by similar forest types and structures.

In this assessment, the impact of a country's NFI capacity (Romijn et al., 2015) on the quality of the values reported to the FRA is illustrated by the statistical parameters reported in Fig. 8 (panels c, d and e). The agreement between country AGB computed from our map and reported in the FRA increased with capacity level (correlation coefficient between 0.46 and 0.91, relative RMSD between 30% and 74%, mean difference between -42% and -12%). Several countries with mean AGB above 200 Mg ha$^{-1}$ and thus in the AGB range prone to underestimation (Fig. 5) were associated with an intermediate NFI capacity level (Fig. 8b), which partly explains the somewhat poor agreement between our average AGB and the average AGB reported in the FRA (Fig.s 8c, 8d and 8e).

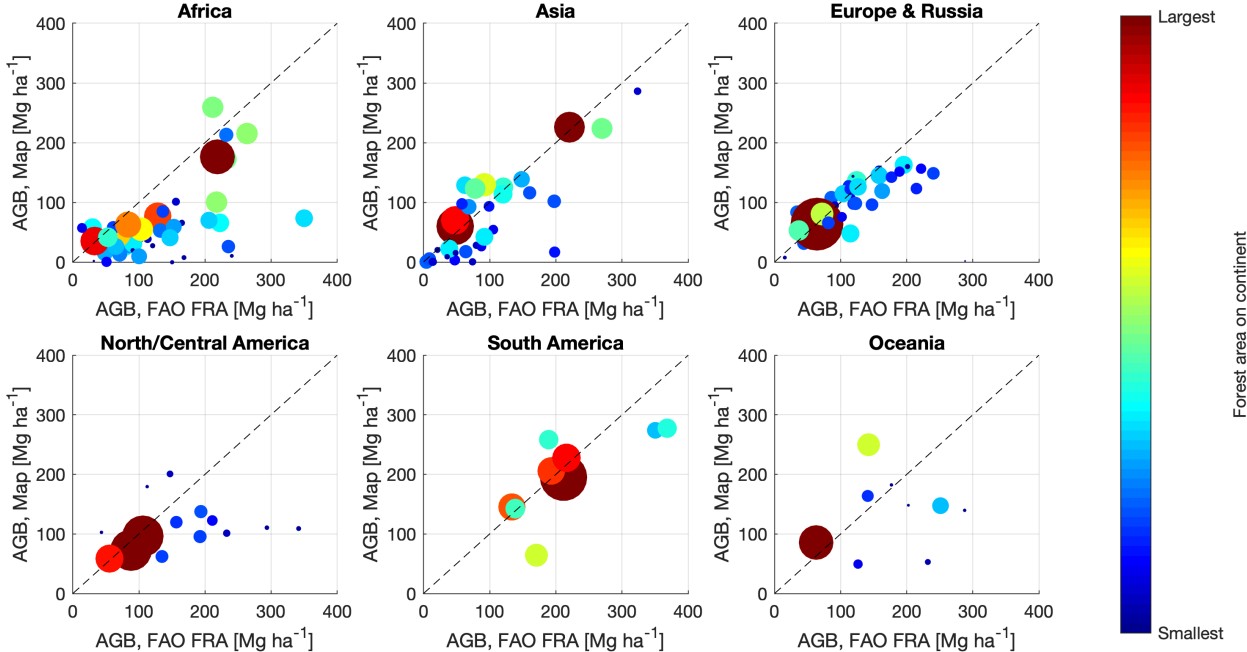

**Figure 7. Average AGB per country from the FAO FRA 2010 country reports and our map dataset. Countries have been grouped per continent. The size of each circle is proportional to the forest area of the country derived from the CCI Land Cover dataset (same scaling across all continents). The colour ramp associated to the circles gives a graphical representation of the relative country forest area (different for each continent).**

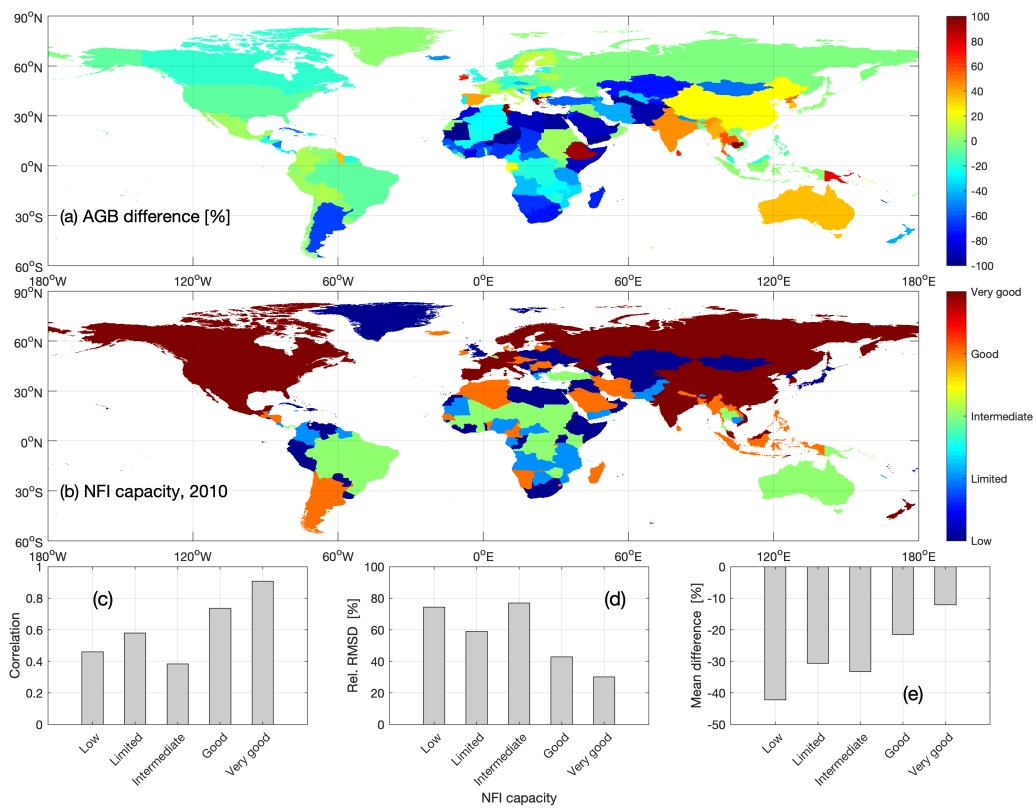

**Figure 8. Difference between country AGB from our map and FRA, expressed relative to the FRA AGB (a), NFI capacities for the year 2010 (Romijn et al., 2015) (b), Pearson's correlation coefficient (c), relative RMSD (d) and mean difference (e) between map estimates of country AGB and values from the FRA for NFI capacity level. Number of countries per NFI capacity level: 51 (low), 30 (limited), 26 (intermediate), 37 (good) and 23 (very good).**

In terms of forest area, the orders of magnitude in the map and the FRA agreed but in most cases the area obtained from the CCI Land Cover dataset was larger than the country values reported to the FRA. The CCI value was on average 24% greater than in the FRA except for the countries of North/Central America, where the average difference was 81%. The discrepancy can be explained in terms of the different definitions of forest used in CCI when generating land cover maps and in the FRA

when using national data of different quality to generate the country estimates. However, the estimate of forest area obtained in this study is likely to be an underestimate, since it excluded land cover classes with a sparse tree and vegetation component that could be attributed to forest under less restrictive definitions of percentage tree cover (Mermoz et al., 2018).

As a result, our country estimates of total AGB were only slightly different from the FRA estimates (Fig. 9) because the lower

AGB densities from our dataset were compensated by the larger forest area values in the CCI Land Cover dataset (Table S7). For Asian and European countries, the difference was 7% and 5% on average, respectively. The difference was larger

throughout the American continent where total AGB estimated in this study exceeded that in the FRA on average by 33% (North/Central America) and 23% (South America). Only for Africa and Oceania did we obtain a total AGB smaller than the FRA, on average -37% for Africa and -29% for Oceania, because of the large difference between average AGB values obtained in this study and those reported by FAO.

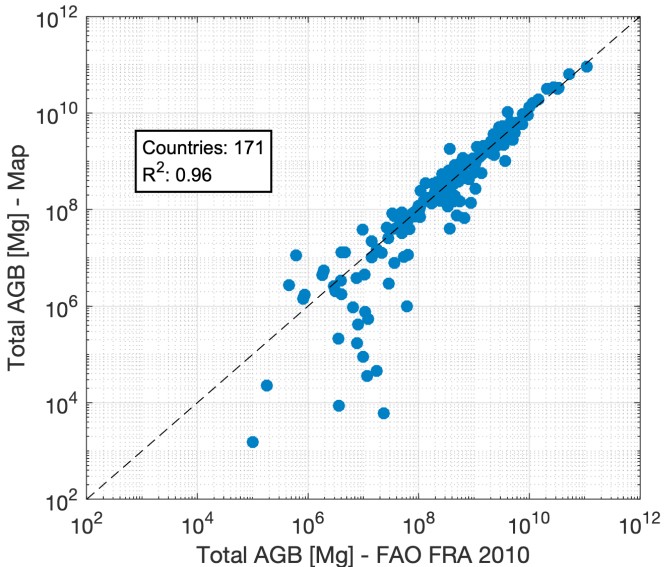

**Figure 9. Total AGB from the FAO FRA 2010 country reports and from our dataset (see Table S7 for details).**

## 3.5 Comparison of map estimates of AGB

All map datasets analysed in this study including ours (Table 3) showed similar latitudinal trends (Fig. 10a), with the highest AGB in the southern hemisphere at 40°S and across the Equator and low AGB in the dry tropics at around 20°S and north of 10°N. However, there were some large differences between the maps (Fig. 10a), illustrating the current uncertainty about the global forest carbon pools.

For the wet tropics between 15°S and 10°N, the AGB from our dataset was close to that from the pan-tropical dataset by Saatchi et al. (Fig. 10b) but is between 10% and 50% lower than the values by Baccini et al. and the GEOCARBON dataset. The analysis of the latitudinal averages based on the 0.1° grid cell AGBs show an overestimate of AGB in these two datasets by about 10% to 60%. On the contrary, the average AGBs from our dataset were only by a few percent values higher than the corresponding values from the plot inventory database (Fig. 11a and 11b).

The differences between our dataset and each of the pan-tropical AGB datasets were much larger in tropical and sub-tropical regions north and south of the wet tropics. In the northern hemisphere, between 10°N and 30°N, Saatchi et al. and Baccini et al. exceeded our estimates by between 40% and 110% (Fig. 9b). For the same latitude range, the GEOCARBON dataset differed from our values by 110% to -47% with increasing latitude (Fig. 10b). The latitudinal averages at the 0.1° grid cells confirmed that all three maps were strongly biased while our dataset was closer to the values obtained from the plot inventory dataset (Fig. 11). The GEOCARBON dataset presented both positive and negative biases, being about 50% of the average AGB from the plot inventory dataset (Fig. 11a). The overestimate exceeded 100% of the average AGB from the plot inventory dataset in the case of Saatchi et al. and Baccini et al. (Fig. 11b). In the southern hemisphere, between 35°S and 15°S, the GEOCARBON dataset and Baccini et al. exceeded our AGB by 20% to 160%, whereas Saatchi et al. differed from our values by between +20% and -60% (Fig. 10b). The comparison at the level of the 0.1° grid cells indicates an overestimate by GEOCARBON and Baccini et al. between 20% and 60% of the average AGB from the plot inventory data (Fig.s 11a and 11b, respectively). The difference between the average AGB from our dataset and from Saatchi et al. with respect to values from the plot inventory dataset were instead mostly below 20%.

For the southern hemisphere temperate forests south of 35°S, our AGB exceeded the values from Saatchi et al. and the GEOCARBON dataset. The comparison was, however, of limited value because of the incomplete coverage by the former and the coarse resolution estimates used to fill gaps in the latter.

North of 30°N, we did not observe differences between our dataset, Thurner et al. and the GEOCARBON dataset. This was expected since all studies were based on the same Envisat ASAR dataset and the BIOMASAR algorithm. The only difference was visible at about 70°N (Fig. 9b and 6c) because of the tendency of ASAR to overestimate biomass in sparse tree vegetation. This difference is, however, of minor importance in the context of global estimation of AGB stocks because the AGB hardly ever exceeded 30 Mg ha$^{-1}$ at these latitudes in any of the datasets (Fig. 10a), this being consistent with values based on forest inventory measurements (Stolbovoi and Mc Callum, 2002; Gillis et al., 2005). Figs 11a and 11c confirm that the three maps present similar spatial patterns of AGB; the difference with respect to the plot inventory dataset was about 20%-30% for all datasets except for GEOCARBON at 60°N where the difference was 60% of the plot inventory value. These results were attributed to the higher spatial resolution of our map. The coarse resolution AGB datasets by Liu et al. and Kindermann et al. agreed with our estimates in the large unbroken tracts of forest in the wet tropics between 10°S and 10°N and in the boreal zone around 60°N (Fig. 10d). However, the estimates of AGB by Liu et al. were up to 70% lower than our estimates in fragmented forest landscapes, e.g. between 50°S and 20°S and between 20°N and 50°N (Fig. 10d). The dataset by Kindermann et al. was better correlated with our estimates, the difference rarely exceeding 30% in absolute terms.

Table 3. Total AGB (Pg) for five latitude ranges roughly corresponding to the temperate forests of the southern hemisphere (60°S - 30°S), the humid and dry tropics of the southern hemisphere (30°S - 10°S), the wet tropics (10°S - 10°N), the humid and dry tropics of the northern hemisphere (10°N - 30°N), and the temperate and boreal forests of the northern hemisphere (30°N and 90°N). Values marked with superscript 1 indicate partial coverage of the latitude range by the corresponding map; n/a indicates not available in the latitude range. The forest area for each range is reported on the last line.

| Source | Latitude range | | | | |
|---|---|---|---|---|---|
| | 60°S - 30°S | 30°S - 10°S | 10°S - 10°N | 10°N - 30°N | 30°N - 90°N |
| This study | 12.5 | 49.8 | 253.9 | 37.3 | 160.8 |
| (Saatchi et al., 2011a) | 4.0 [1] | 51.8 | 253.7 | 69.0 [1] | 13.1 [1] |
| (Baccini et al., 2012) | n/a | 55.8 [1] | 278.8 | 39.4 [1] | n/a |
| GEOCARBON (Avitabile et al., 2016) for the tropics, (Santoro et al., 2015a) for the northern hemisphere | 10.3 | 39.9 | 248.0 | 19.3 | 108.2 |
| (Thurner et al., 2014) | n/a | n/a | n/a | n/a | 114.7 |
| (Liu et al., 2015) | 9.6 | 58.6 | 278.6 | 53.1 | 169.3 |
| (Kindermann et al., 2008) | 13.1 | 74.8 | 242.8 | 48.1 | 175.8 |
| Forest area ($10^6$ ha) | 109.3 | 584.0 | 1366.1 | 497.8 | 2294.7 |

All AGB datasets yielded similar values of total AGB in the wet tropics between 10°S and 10°N (Table 3) with our estimate being intermediate to the others. For the latitude ranges corresponding to the humid and dry tropics, our total AGB was much lower than Saatchi et al. and Baccini et al. (87.1 Pg vs. 120.8 Pg and 95.2 Pg, respectively) because of their substantially higher estimates of AGB outside the rainforest region. It was also different from Liu et al. and Kindermann et al. (87.1 Pg vs. 111.7 Pg and 122.9 Pg, respectively) because the coarse spatial resolution of the data did not allow forest fragmentation to be accounted for. For boreal and temperate forests (Table 3, column: 30°N - 90°N), the total AGB from our dataset was larger than in Thurner et al. and the GEOCARBON dataset (160.8 Pg vs. 114.7 Pg and 108.2 Pg, respectively). In both these datasets, a more stringent definition of forest than the CCI Land Cover dataset was used resulting in a large number of AGB estimates equal to 0 Mg ha$^{-1}$ in areas labelled as forest by the CCI Land Cover dataset. Our estimate of total AGB was slightly lower

than that from Liu et al. and Kindermann et al. because these datasets did not account for forest fragmentation. The values were difficult to compare for the latitude range including the temperate forests of the southern hemisphere (Table 3, column: 60°S - 30°S) because of the approximations underlying the AGB values obtained in the other maps (extrapolation from coarse resolution estimates in GEOCARBON, statistics of large countries in Kindermann et al. and use of inversion models trained in the tropics in Liu et al.).

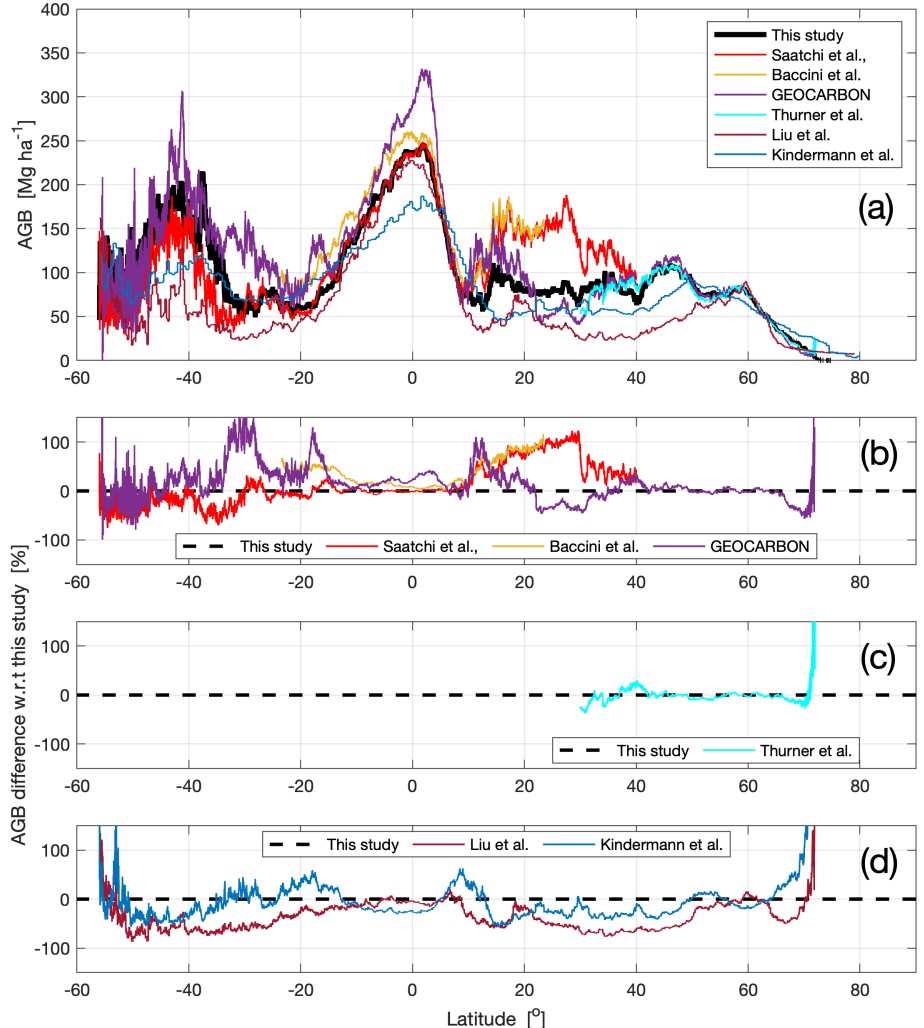

**Figure 10. Profiles of average AGB along latitude for each dataset listed in Table 3 (a), together with the corresponding relative difference between the latitudinal AGB derived from the other datasets and this study (b, c and d), expressed in percentage values relative to our estimates.**

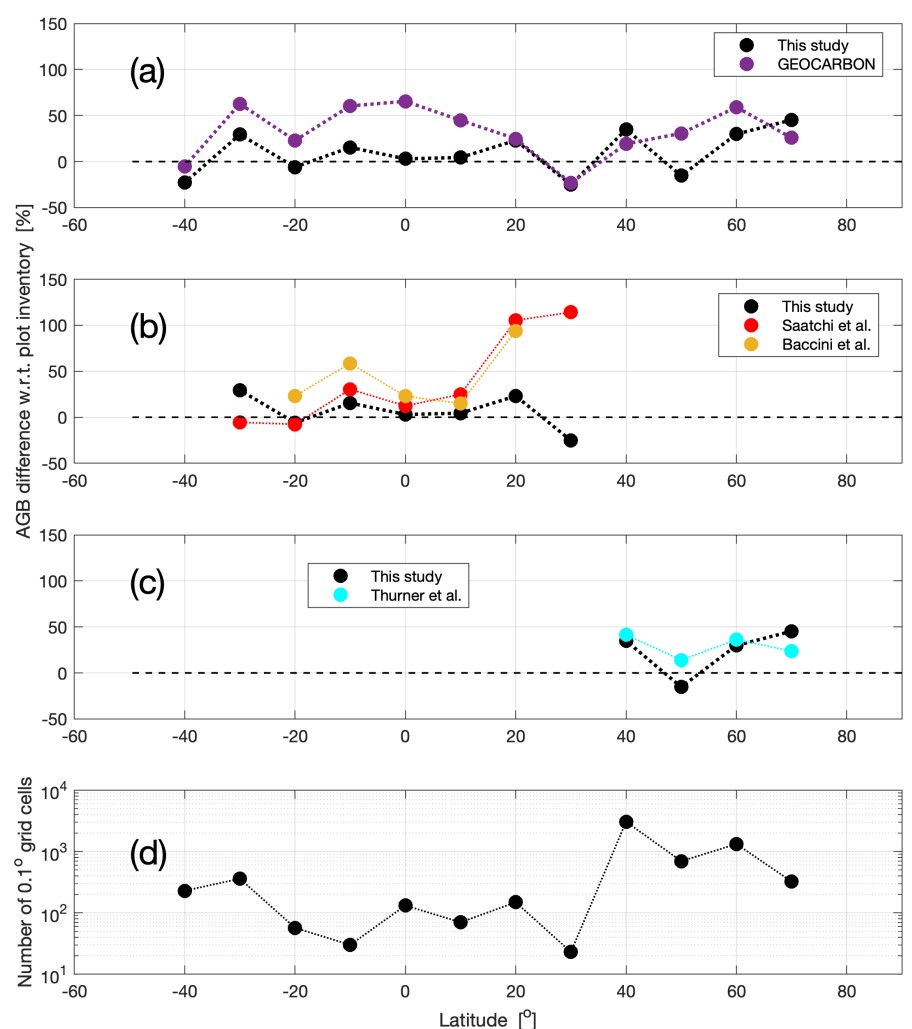

**Figure 11. Relative difference of AGB from maps and plot inventory data averaged over the 0.1° grid cells validation cells (a, b and c). The number of grid cells stratified by 10°-wide latitude intervals is illustrated in (d).**

## 4 Discussion and conclusions

Overall, the results indicate that our maps reproduced known spatial patterns of AGB (Fig. 5, Fig. S12 and Fig. S13) and GSV

(Fig. S14) correctly, although with some systematic errors (Fig. S12, Table S8). Independent assessments confirmed these

indications (Spawn et al., 2020; Schepaschenko et al., 2021). The dispersion of the AGB estimates about the identity line in

Fig. 5 was often explained by regional biases (Fig. S12) arising from approximations in the retrieval model and the use of

remote sensing data with different spatial resolutions (Table S8). Hence, the global RMSD and bias statistics in Fig. 5c had

limited informative value when trying to explain our results. Instead, values at the level of continents or domains provided a

more realistic indication of the quality of the AGB dataset since we could identify a predominant error source in each (Table S8).

Interpreting Table S8, errors were partly a consequence of a simple model relating SAR observations to biomass through three unknown parameters, whose estimates relied on several assumptions and generalization rules. In addition, to avoid unrealistic model fits when relying on external data for model training, our model training procedure was forced to follow the average relationship between SAR backscatter and biomass. This reduced the variability of the predicted backscatter with respect to the measured backscatter, which translated to overestimation in the low AGB range and underestimation in the high AGB range, as evidenced by Fig. 5. In spite of these weaknesses, we preferred such an approach to the traditional model training based on a dataset of reference biomass measurements because of the paucity of reference measurements available globally, which would lead to major AGB biases in regions not represented in the training dataset.

Underestimation of AGB in dense mature forests (Fig. 5 and Fig. S9) were further exacerbated by the weak sensitivity of SAR backscatter to forest variables (Table S8). The range of AGB beyond 150 Mg ha$^{-1}$, in particular in wet biomes, corresponded to a backscatter range at C- and L-band of less than 0.3 dB (Cartus and Santoro, 2019; Santoro et al., 2015a) and thus had substantial uncertainty. Combining estimates of biomass from multiple C-band Envisat ASAR observations increased the accuracy of the AGB retrieval compared to individual estimates (Santoro et al., 2011). However, the ALOS PALSAR L-band dataset consisted of a single observation so that any contribution to the SAR backscatter not related to the forest itself translated into a bias. This was most relevant in the wet tropics where AGB was estimated solely on such single observations. Imperfect modelling of the ALOS PALSAR backscatter in sloping terrain caused an additional bias, which propagated through to the estimate of AGB (Fig. S14).

Characterization of the spatial variability of wood density and biomass expansion factors was critical. We acknowledge that in less sampled regions, such as savannah, the estimates of AGB might have been limited by the observations used to estimate WD and BEF. Nonetheless, even if we could not quantify the impact of local BCEF biases on the AGB estimates because GSV observations were unavailable for most of the inventory, our understanding of the retrieval errors (Table S8) and uncertainties (Fig. S10) indicate that the main contributors to the errors were not associated to the BCEF layer. This interpretation, however, is affected by the opportunistic dataset of plot inventory measurements available to assess the AGB estimates. We agree that further research opportunities should focus on understanding the implications of local wood formation strategies to AGB estimates at local scales and across environmental gradients.

Our estimate of the total AGB in forests increased from 522 Pg to 596 Pg when further accounting for the woody biomass in non-forest CCI land cover classes. An additional 4 Pg was estimated in cropland and non-vegetated land cover types, resulting in a terrestrial woody AGB pool of 600 Pg, corresponding to approximately 282 PgC assuming the mean carbon fraction in

woody vegetation of 0.47 (IPCC, 2006). Based on our dataset and accounting for additional maps of non-woody biomass and auxiliary datasets, a more comprehensive report of the global carbon stock both above- and below-ground was recently published (Spawn et al., 2020). The lack of AGB estimates for the Western Pacific Islands had a negligible effect on our total AGB values. AGB underestimates and substantial uncertainty in our dataset arose from having only a single observation of L-band backscatter and the inability of C-band observations to resolve levels of AGB in forests with high biomass stocks. This effectively led to use of a single radar observation to estimate biomass in dense tropical forests, causing our values to be at the low end of the range of estimates of the terrestrial AGB pool (Table 3 and Table S1). Although our estimates for the wet tropics were affected by bias, our dataset provides new insights into the spatial distribution and levels of AGB in tropical, sub-tropical and temperate forests (Table 3, Fig. 10) and provides overall more accurate spatial patterns of the terrestrial carbon pool than previous maps (Fig. 11). For the boreal domain, our estimates agreed with previous maps, which were reported to be unbiased (Table 3, Fig. 10), whilst providing a more detailed portrait of the spatial distribution of AGB (Fig. 3) because of the higher spatial resolution of the remote sensing datasets used in this study (1 ha vs. 100 ha).

At country level, we demonstrate the benefit of our map-based estimates of AGB and total AGB in the context of global reporting of biomass pools (Table 2, Fig. 7). Nonetheless, our dataset may not be sufficiently accurate for individual countries to use it as the only basis for reporting biomass and carbon pools, in particular for countries where the AGB estimates may be experiencing systematic retrieval errors (e.g., predominant high AGB, limited sensitivity of the backscatter to biomass, strong topography). However, it may provide support to national reporting in conjunction with a forest inventory system (Næsset et al., 2020). Our dataset agreed well with values reported in the 2010 FRA for countries with an established national forest inventory (Fig. 7, Table S7). However, for countries reporting AGB on the basis of expert knowledge, best guess estimates and default parameters, we identified clear regional patterns of either overestimation (e.g., in Africa, see Fig. 7) or underestimation (e.g., in Asia, see Fig. 7), which could be related to the capacity of national forest inventories (Fig. 8). Although the global average AGB in our map differs from the FRA value by only 8% (108 Mg ha$^{-1}$ vs. 119 Mg ha$^{-1}$, Table 2), the smaller forest area reported in the FRA implies that our total AGB estimate exceeded the value reported in the FRA by 11%.

Our dataset provides new insights on the spatial distribution and magnitude of terrestrial woody AGB as well as a valuable resource for climate and carbon modelling because of its completeness, standardized estimation procedure and high level of detail. The major caveat of past and current spaceborne remote sensing observations used to estimate biomass is their limited sensitivity to any forest parameter. Such limitations may be partly overcome by combining measurements and models, as demonstrated in this study. Nonetheless, it will only be during the 2020s that observations from space will allow accurate quantification of the terrestrial biomass pool. The availability of a wide range of observations from space, including LiDAR (Dubayah et al., 2020) and P-band SAR (Quegan et al., 2019) are expected to provide more detailed information on the vertical

structure of forests, thus adding up to the information by backscatter measurements sensitive mostly to horizontal properties and thus of limited reliability in tall and dense forests.

**Data availability**

The AGB and GSV datasets can be downloaded from http://globbiomass.org/products/global-mapping or from the PANGAEA repository at https://doi.org/10.1594/PANGAEA.894711 (Santoro, 2018).

**Author contributions**

M.Sa. and O.C. designed the study, developed the GSV retrieval algorithm, generated the GSV and AGB datasets, and interpreted the results. N.C. designed the GSV to AGB conversion and generated the conversion dataset. D.R., V.A., A.A.,

S.d.B. and M.H. designed the validation procedure of the AGB dataset, compiled the database of plot inventory data and computed the grid cell values from the reference database. S.Q. acted as scientific lead of the study. P.R.V, H.B. and J.Car. contributed to the validation of the AGB dataset in South America and Africa. D.S. and M.K. contributed to the validation of the GSV dataset in Russia. M.Sh. and T.I. generated the ALOS PALSAR mosaics and provided support to their use. A.M.M. contributed to the estimation of the wood density dataset. J.Cav., R.C.G., P.d.C.B., N.D., N.L., J.Lia., J.Lin., E.T.A.M., A.M.,

840     A.P.P., C.M.R., F.S., G.V.L., H.V., A.W. and S.W. provided forest plot data. M.S. and O.C. wrote the manuscript. S.Q. overviewed the writing of the manuscript. All authors contributed to the drafting of the manuscript.

**Competing interests**

The authors declare they have no competing interests.

**Disclaimer**

The data is provided as is with no warranties.

**Acknowledgments**

The work was supported by the European Space Agency (ESA) within the Data User Element (DUE) GlobBiomass project (ESRIN contract No. 4000113100/14/I-NB). The National Centre for Earth Observation was supported by the UK Natural Environment Research Council. The FOS data collection for Russia was performed within the framework of the state assignment of the Center for Forest Ecology and Productivity of the Russian Academy of Sciences (no. AAAA-A18-118052590019-7), the Russian ground data preparation and pre-processing were financially supported by the Russian Science

Foundation (project no. 19-77-30015).

We are thankful to the GlobBiomass project team and F. M. Seifert (ESA) for valuable suggestions and stimulating scientific discussions. We are thankful to T. Tadono (JAXA EORC), M. Hayashi, (JAXA EORC), K. Kobayashi (RESTEC), A. Rosenqvist (soloEO) and J. Kellndorfer (EBD) for support with the use and interpretation of the ALOS PALSAR mosaics.

Support by the CCI Land Cover project team, in particular, S. Bontemps, UCL, is greatly acknowledged. The help from Martin Jung (MPI-BGC) in feature selection and Ulrich Weber (MPI-BGC) for data processing for the GSV to AGB conversions is greatly acknowledged. Forest inventory data for the validation of the AGB map were made available among others by Ben de Jong, the Prince Edward Island Department of Communities, Land & Environment, Forests, Fish & Wildlife Division and the Nova Scotia Department of Natural Resources. Inventory Data were also provided by the Sustainable Landscapes Brazil project supported by the Brazilian Agricultural Research Corporation (EMBRAPA), the US Forest Service, and USAID, and the US Department of State. We thank the two anonymous reviewers, contributors to the short comments, G. Hengeveld and J. Chave for reviewing and improving the manuscript.

Envisat ASAR data have been distributed and processed on ESA's Grid Processing On Demand (G-POD) platform under ESA's Category-1 Project IDs 6397 and 9209. Access to G-POD was obtained under ESA's Category-1 project nr. 9209. R. Cuccu, J. M. Delgado Blasco, S. Pinto and J. Farres, ESA/ESRIN, are acknowledged for implementation of SAR processing chain on G-POD and support.

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
