# Peer review of "The global forest above-ground biomass pool for 2010 estimated from high-resolution satellite observations"

_Earth System Science Data, 2020_

## Short Comment (SC1) · 10 Nov 2020

Our team analyzed this product in the scientific project and obtained good results in comparison with ground (field) data. We are glad that a comprehensive article with a description of the methodology and validation has appeared.

---

## Short Comment (SC2) · 14 Nov 2020

This paper is of high interest due to its strong uncertainty analysis and reliable size/quality of the training/validation data. However, since Ploton et al. (2020) paper was published (https://www.nature.com/articles/s41467-020-18321-y), there is an explicit need to consider spatial patterns in the training data. Typical situation when predictions at pixel level are too differing from the independent validation data can be a legacy of such a spatial autocorrelation issue. It would be interesting to understand whether this data is sensitive to the somewhat hidden biases provided by the location of sample plots. Good luck for authors of the paper to supplement with a new global

dataset which can provide more reliable predictions at some scales and locations compared to the existing similar datasets.

---

## Short Comment (SC3) · 2 Dec 2020

Thanks for an interesting manuscript. Very important work.

Noting the importance of the L-band SAR data in the development of the above-ground biomass map, I'd be interested to know the authors' (or any other experts) opinion on how data from current and near-future L-band SAR missions could be further enhanced to better accommodate information extraction for this kind of applications.

We know that increased temporal revisit is of key imporance to mitigate seasonal and weather-related effects affecting the data. That parameter is also well acknowledged

by space agencies and addressed in near-future missions (e.g. NISAR).

When it comes to polarisation, today's SAR missions are not taking advantage of the full SAR capacity. It is widely known that both the co-pol and the cross-pol channels provide critical and complementary information about vegetation structure, which is why co/cross-pol is the preferred dual-pol (DP) mode for today's key missions such (PALSAR/PALSAR-2 and even Sentinel-1). But does DP provide the whole picture?

I'd be interested in hearing some opinions on the actual usefulness of fully polarimetric (qual-pol, QP) L-band data for applications related to forest and biomass. QP data provides detailed information about the scattering mechanisms, which is potentially of great relevance for vegetation biophysical parameter retrieval. It also accommodates corrections for Faraday rotation. It is however notable that also forthcoming missions such as NISAR and ROSE-L have ditched QP in favour for DP only. While DP was a given choice in the ALOS/ALOS-2 era, where the the narrow QP swath width was a major constraint that in practice prevented systematic global observation at QP, it is not necessarily the case for the next generation missions. Although the QP swath width still is typically half of that for DP (hence influencing the effective temporal revisit frequency), the next generation missions all operate with very large swaths that would allow systematic QP acquisitions with monthly repeat (e.g. 28 days in the case of ALOS-4).

So coming back to my question – would global/regional Pol-SAR/Pol-InSAR observations bring any new information of relevance for forest structure and biomass measurements? Or are the advantages so marginal that QP would simply be considered a waste of satellite resources? Any thoughts?

---

## Short Comment (SC4) · 4 Dec 2020

Across many savanna regions (Africa, Australia, Brazil (cerrado)), there are a number of environmental gradients (e.g., temperature, precipitation, evapotranspiration) that impact on the allocation of biomass to both the above and below ground components. The species diversity also varies and these savannas are, in some areas, dominated by coniferous rather than broadleaved species, which allocate biomass differently. There are also variations according to long term environmental conditions (e.g., drought, persistent fires). Over large areas in these environments, one or just a few Biomass Expansion Factors might be applied and this may lead to errors in the biomass esti-

mation. It seems that there is potential for more studies to link the BCF to a range of environmental controls and thereby provide a more targeted assessment of BCF and concurrently improve the estimates of AGB. A similar approach might be undertaken for wood density. This research will advance as we move forward regardless because of requirements and interests but there are opportunities for studies to be put in place earlier given the global need for biomass information.
* * *

---

## Short Comment (SC5) · 4 Dec 2020

The paper provides a very good assessment and comparison with other global products. These are likely to increase in number and quality given the increased capacity for generation as a result of increased amounts and diversities of data from satellite sensors operating in different modes (optical, radar, lidar), many of which have specified mission focus on biomass (e.g., GEDI, BIOMASS, NISAR). The maps generated will be used for different purposes (e.g., climate modelling, national reporting of carbon stocks and change). For this, users will need to make decisions on whether to use these maps and, if so, which is the most reliable. In the GlobBiomass product,

error assessments are robust and can be used to support decisions. However, other maps including updates of the product presented might not have such rigour or may approach in a different way. The use of a standardized global ground dataset with which to compare all products is recommended but there are complications in terms of the timing of these relative to that of the satellite observations and also how these are summarised and over what time periods. It would be useful to provide advice for potential users to help make decisions on how, where and when to use these different maps.

---

## Short Comment (SC6) · 4 Dec 2020

This is a study may become a good reference for biomass studies in forest areas with low inventory capacity. The assessment of previous AGB remote sensing maps is also an opportunity to improve the estimation of potential AGB using satellite images.

---

## Short Comment (SC7) · 4 Dec 2020

Our team compared the growing stock volume from the GlobBiomass map against 1800 in-situ plots from the Romanian Forest National Inventory and using 104 large (1 ha) sites. For the NFI data, the comparison was carried out at both, Permanent Sample Plot - PSP level (500m2) and at NFI Grid node level (i.e., using the averaged PSP volume over the 6ha corresponding to each grid node). A grid node is the center of a square (250m x 250m) with 4 PSP as vertices.

We observed a large discrepancy between the European FRA based assessment (-8%, Paragraph 585) and our results regardless of the method used (i.e., NFI PSP, NFI

grid node, Large sites). The difference between the mapped values and the in-situ data reached RMSE% errors between 50%-70%. The map sub-estimates the in-situ volumes by 30 m3 (oak forests) to 345 m3 (beech forests).

It would be interested to understand/discuss to what extent the GlobBiomass map may be used to estimate regional or national stocks over biomes that support high biomass values.

---

## Short Comment (SC8) · 4 Dec 2020

One line 270. It references a "4 x 4 window" for spatial averaging. Hopefully a simple query in how that worked, normally window based averaging would use an odd number sized window (i.e., 3x3, 5x5).

It would be interesting to consider whether mangrove AGB may be compromised by forest structure and inundation patterns, which does tend to reduce the AGB estimates from L-band SAR. Furthermore, do the biomass expansion factors consider root biomass considering this can be quite a large component.

---

## Author Comment (AC1) · 13 Jan 2021

We are thankful for this analysis and the comment to the manuscript. Your team's contribution to the assessment using additional data not available in our AGB measurement database is highly appreciated.

---

## Author Comment (AC2) · 13 Jan 2021

This comment is welcome because it sheds light on a feature of potential positive impact on global biomass estimation that could not be exploited to generate the dataset presented in this manuscript. Physically, the availability of a fully polarimetric dataset provides the most complete description of the forest structure seen by radar. Conversely, a dual-polarized acquisition misses part of the forest structure. The question, however, is which radar observable makes the difference.

In terms of radar backscatter, as used in this study, a fully polarimetric dataset is in our opinion of marginal benefit compared to a dual-polarimetric dataset. This is probably

the reason why fully polarimetric observations are usually ditched. Radar backscatter is a simple observable compared to more advanced observables that can be obtained from radar observations, e.g., through polarimetric, interferometric or tomographic processing. Nevertheless, these advanced observables are necessary for vegetation studies. The real potential of the fully polarimetric dataset is in the phase component of the signal because the phase captures the spatial variability of the vegetation structure more than the intensity. At L-band the scenario is, however, tricky because part of the vegetation is transparent, so the understanding of the signal is not straightforward.

Probably, the lack of a broad understanding of how the diversity of vegetation structure globally affects the polarimetric signal is the answer to the questions above. The global forest biomass retrieval presented in this study is based on evidence from an extensive literature (Santoro and Cartus, 2018) that assessed radar backscatter observations all over the world. This understanding is, in our opinion, still in its infancy for polarimetric observables. The lack of repeated observations at multiple locations is a reason for the slow development of large-scale thematic mapping based on polarimetric observations. Given the lack of knowledge, it is therefore premature to advocate satellite missions focusing on fully polarimetric observations. Of course, other variables affect the decision whether to select a DP over a FP mission (e.g., revisit times, bandwidth, frequency of observations) but this goes beyond the question posted in this comment on whether FP observations are of benefit to global forest biomass estimation.

Narrowing down our reply to this study, the availability of a global dataset of L-band FP data consisting of multiple acquisitions throughout one year would have been of enormous benefit to overcome what we see as two major weaknesses, i.e. the underestimation in forests with high AGB and the inability to capture small-scale variability of vegetation structure.

In our retrieval procedure, GSV (and thereof AGB) is predicted from observations of the radar backscatter, which combines horizontal properties of the canopy (density of trees, branches, foliage) and vertical properties (height) in a single measurement. The

limited capability of C- and L-band signals to penetrate dense canopies leads to weaker sensitivity of the backscatter to taller and denser forests. Resolving such forests with a closed canopy and different heights, i.e., different biomass levels, with backscatter observations implies considerable errors. In our study, several measures were implemented to reduce these errors (multi-temporal observations, spatially adaptive estimation of model parameters, simplifications in the retrieval models, etc.). Nonetheless, we could not overcome the inherent limitations of the observations, resulting in systematic retrieval errors

Again, the fact that the radar backscatter is an ensemble of horizontal and vertical properties of the vegetation is a limiting factor in ecologically diverse environments. Although we did not assess the global AGB dataset with respect to ecological zoning, we may assume that the dataset presents some systematic issues in regions where the radar observables could not resolve different vegetation structures. Resolving structures by polarimetric observations would be of substantial aid to improve biomass mapping.

In conclusion, although this manuscript does not discuss which satellite observations are most suited to estimate biomass globally, this comment provides some indications on the role of FP observations in future biomass mapping activities: 1. Foster studies on polarimetric observables at multiple sites; such studies should assess simultaneously the benefit of multi-temporal and multi-frequency observations 2. Undertake benchmark studies of FP vs. DP-based retrievals in forests prone to retrieval errors with the latter type of data (tropical rainforest, ecologically diverse environments) 3. Explore whether the polarimetric observables from global FP datasets (ALOS-1 and ALOS-2) present a diversity of values compared to the basic radar backscatter from DP observations

A solid knowledge of the FP observables would certainly foster discussion on the gain from implementing FP observations in future L-band SAR missions.

**Reference**

Santoro, M. and Cartus, O.: Research Pathways of Forest Above-Ground Biomass Estimation Based on SAR Backscatter and Interferometric SAR Observations, Remote Sensing, 10(4), 608, https://doi.org/10.3390/rs10040608, 2018.

———————————————

---

## Author Comment (AC3) · 13 Jan 2021

Global biomass maps, such as the GlobBiomass dataset, try to address the demand for better knowledge of the distribution of biomass pools. By benchmarking the Glob-Biomass dataset against the FAO FRA statistics, we have identified regions where the map improves current estimates that are based on only a few measurements and default reference values. By cross-comparing existing data products with an extensive database of plot inventory measurements (even if opportunistic), we provided indications of where knowledge about the biomass distribution on Earth is most uncertain. This is a substantial advance embodied in our dataset and provides potential users

with a guide to the reliability of the biomass spatial patterns reproduced by currently available global datasets.

This paper also notes that the GlobBiomass dataset has deficiencies that were identified by comparing against in situ observations. For regions where such data were available, we were able to relate errors to sub-optimal remote sensing data to estimate biomass, simplified models, assumptions and generalizations, etc. In most cases, errors did not arise from a single cause, as shown by the regional scatterplots comparing estimated and reference AGB (Figure 3). For this reason, we refrained from providing global error statistics (e.g., root mean square error). However, the in situ dataset is opportunistic and we may have missed regions with considerable errors, so a comprehensive standardized global ground dataset would be extremely valuable in providing a more complete assessment.

From these observations, it is clear that the provision of advice to potential users on how to use current global biomass maps has multiple facets. It is clear that stronger interaction between map producers and users would is needed (a) to better understand user requirements and criteria that must be met for them to find the products useful, and (b) to provide guidelines to users on which aspects of the data set can be treated as reliable and which contain pitfalls (see e.g., the replies to SC1, SC6 and SC7).

Beyond addressing the needs of the users, maps such as the GlobBiomass dataset set the stage for future mapping endeavours, which will use more robust retrieval methods and, more importantly, rely on a wider range of observations from space capturing different aspects of "biomass". As a result of this comment, we have added a note in the Conclusions of the manuscript.

---

## Author Comment (AC4) · 13 Jan 2021

We are thankful for this comment, which is in line with our understanding on how to provide advice to potential users of the GlobBiomass dataset (see reply to SC5).

---

## Author Comment (AC5) · 18 Jan 2021

The prediction of GSV from the SAR data was not based on a dataset of training samples. Instead, it relied on a self-calibration approach, which was referred to on lines 220-222. Thereafter (lines 223-227), we explain briefly the reasons for not training the retrieval models with in situ measurements. The rationale and implementation of model training are presented in Sections A.2 and A.3 of the Supplement. In conclusion, the issue of spatial autocorrelation does not apply to our predictions.

---

## Author Comment (AC6) · 18 Jan 2021

We completely agree that the spatial variability of wood density (WD) and biomass expansion factors (BEF) is indeed critical and we acknowledge that in less sampled regions, like savannas, the estimates of AGB may be limited by the observations used to estimate WD and BEF. Patterns of biomass allocation between stem to total AGB should be different in drier regions, where plant investment would favour structural organs (stem versus leaf), but this difference would be substantially smaller than the above-to-below ground differences as is suggested. However, these above-to-below ground differences do not enter our estimates that focus on AGB retrievals. Figure S9

shows that the contribution to the AGB standard deviation from the WD and BEF layers was between 25% and 40% in dry tropical environments, while Figure S19 shows that the WD estimates are unbiased at coarse scales across ecological zones. These results reflect the limitation in capturing the local variability in WD, and BEF, which may result from the limited observations, from the spatial mismatch between in situ data and covariates, or from the natural variability that is not captured by environmental factors. However, limited observations in dry tropical forests may imply a stronger generalization of the models used to predict WD and BEF in these regions and justify the slight underestimation of WD in these regions (between 0.05 and 0.1, Figure S19). Overall, the models to predict WD and BEF were developed to be globally unbiased (see Supplement A.5). This ensured that ensemble averages would be correct while accepting that some the predictions of the BCEFs (biomass conversion and expansion factor, i.e., the product of WD and BEF) could be locally erroneous.

Although we could not quantify the impact of local BCEF biases because GSV observations were unavailable for most of the inventory, our understanding of the retrieval errors (Table S8) indicates that the main contributors to the errors were the weak sensitivity of the input observations to biomass or the simplifications in the retrieval model. This interpretation, however, is affected by the opportunistic dataset of plot inventory measurements available to assess the AGB estimates. We agree that further research opportunities should focus on understanding the implications of local wood formation strategies to AGB estimates at local scales and across environmental gradients.

We acknowledge that the impact of the BCEF on the dataset was not discussed in the original manuscript and have added a short paragraph in Section 4 accordingly.

---

## Author Comment (AC7) · 18 Jan 2021

We welcome this independent assessment of the GlobBiomass dataset because Romanian forests were not represented in the plot database. We can explain the large RMSE as a consequence of (i) high AGB in large parts of the country, i.e., in the range where AGB in Europe was often underestimated (Figure 3), (ii) the strong topography of the Carpathian mountains and (iii) the relatively low value of maximum biomass set for this region (note: this was recently recomputed based on additional datasets not available at the time of the study). This interpretation of the errors is in line with interpretation of errors in Table S2 for regions with high carbon stocks. Our understanding

is that in such regions, the average AGB derived from the GlobBiomass dataset may be an under-estimate; the difference depends on the region, as errors are a consequence of the interplay of multiple factors. For this reason, it is not possible to provide an AGB level above which individual map values should be considered as less reliable. We have added a note in Section 4 to highlight that not everywhere average values (e.g., for countries) have the same reliability.

Nevertheless, in a recent paper, Naesset et al. (2020) demonstrate how global biomass maps locally calibrated on in situ plots can increase the precision of local stock esti­mates computed from only the plot data.

References

Næsset, E., McRoberts, R.E., Pekkarinen, A., Saatchi, S., Santoro, M., Trier, O.D., Za­habu, E., and Gobakken, T., Use of local and global maps of forest canopy height and aboveground biomass to enhance local estimates of biomass in miombo woodlands in Tanzania, International Journal of Applied Earth Observation and Geoinformation 93. 2020. https://doi.org/10.1016/j.jag.2020.102138.

―――――――――――――――

---

## Author Comment (AC8) · 18 Jan 2021

1) The original pixel size was equivalent to a square of approximately 25 m on each side on the ground. Averaging to 1 hectare implied a 4 x 4 window size. The area covered by the 16 pixels within 1 hectare was associated with the average AGB from the 16 pixels.

2) Unlike most forests, where backscatter increases with increasing biomass, the relationship for mangroves can be substantially different. In Figure SC1, we illustrate ALOS PALSAR backscatter observations as a function of mangrove height, defined as the elevation of mangroves in the Shuttle Radar Topography Mission (SRTM) C-band

dataset. Assuming AGB is directly proportional to canopy height, it is reasonable to assume that the same patterns apply to the relationship between SAR backscatter and AGB. At several locations we observed lower backscatter in dense mangroves than in surrounding mangroves with lower biomass. This cannot be reproduced by the Water Cloud Model used in our study, since the model always gives higher canopy backscatter with increasing; hence inversion of the Water Cloud Model assigned low biomass to dense mangroves.

One way to overcome this problem would be to implement mangrove-specific training of the Water Cloud Model; however, this runs the risk of an erroneous estimate of the model parameter representing the ground backscatter. Alternatively, one could rely on other sources of remote sensing data, such as interferometric height proposed by Simard et al., 2019. Unfortunately, a global interferometric dataset was not available for the 2010 decade at the time of study. In conclusion, we do not see an optimal global solution for improving the AGB estimates for mangroves and suggest that the GlobBiomass AGB over mangrove areas is used with caution because of potential underestimates.

This discussion for mangroves applies to any forest structure in principle. While some are modelled best with the proposed Water Cloud Model (e.g., boreal forests), some others may suffer systematic errors due to the simplified parameterization of the model. The choice of the model is, however, subordinate to the observations available. A few observations of the radar backscatter at C- and L-band limit the performance of a retrieval model, whatever its formulation may be because of the vertical structure of the forest is not captured in its full extent. Observations sensitive to forest structure from spaceborne laser or longer wavelength radar will overcome this issue. This aspect was briefly introduced at the end of the manuscript and has been revisited in the revision in order to duly account for this comment and have it discussed from a more general perspective.

3) The BEF only refers to above-ground biomass, i.e. expansion from stem biomass to
total above-ground biomass density. We have added above-ground in the manuscript to avoid confusion.

References

Bunting, P., Rosenqvist, A., Lucas, R., Rebelo, L.-M., Hilarides, L., Thomas, N., Hardy, A., Itoh, T., Shimada, M. and Finlayson, C.: The Global Mangrove Watch—A New 2010 Global Baseline of Mangrove Extent, Remote Sensing, 10(10), 1669, https://doi.org/10.3390/rs10101669, 2018.

Simard, M., Fatoyinbo, L., Smetanka, C., Rivera-Monroy, V. H., Castañeda-Moya, E., Thomas, N. and Van der Stocken, T.: Mangrove canopy height globally related to precipitation, temperature and cyclone frequency, Nature Geoscience, 12, 40–45, 2019.

**Fig. 1.** Median backscatter as a function of Shuttle Radar Topography Mission (SRTM) elevation for five locations labelled as mangroves by the Global Mangrove Watch classification (Bunting et al., 2018).

[Figure]

---

## Short Comment (SC9) · 4 Feb 2021

I have explored this datasets extensively and find it to be of an excellent quality. The manuscript is also very well written and clear. The authors might consider mentioning a recent study out of the University of Maryland (see: https://doi.org/10.1088/1748-9326/ab2917) that compared their Globbiomass product to a very high resolution biomass map produced specifically for the northeastern region of the United States. The UMD authors find good agreement between the two products in their focal region– better agreement, even, than they find between two national maps based exclusively on optical imagery (those of Blackard and Wilson).

---

## Referee Comment (RC1) · Anonymous Referee #1 · 7 May 2021

Santoro et al eloquently describe their global maps of [woody] forest biomass in a way that is both thorough and approachable. Unlike previous biomass maps, they leverage global SAR composites to first generate a map of growing stock volume, from which they then generate an additional map of the biomass density. As I understand it, this approach contrasts with those used to generate previous forest biomass maps which have instead used more direct approaches that are thus more correlative and perhaps, then, less reliable. Santoro and colleagues also generate an uncertainty map, making their product distinct from its predecessors in that regard; I particularly appreciate the transparency this affords the product. Overall, I commend the authors for a well written

paper and robust product.

I am admittedly not an expert on non-optical remote sensing and would thus characterize my feedback below as more representative of a potential end user. While I can't speak much to the methods Santoro et al use, I do feel this perspective is important given the number and breadth of biomass maps that have been published in the last 10+ years. Below, I have identified several minor issues that I feel should be addressed to ensure that users like myself properly understand and interpret these data. Some are inevitably a matter of opinion and I respect that the authors may feel differently. Overall, though, I think this manuscript and the data it described will (at last!) be great contributions to our field.

Minor Comments:

Line 87 – Baccini et al. (2017) would seem to be an appropriate reference for forest degradation.

Line 85 – Gibbs et al. (2007) would seem to be an appropriate reference for REDD.

Line 86 – You might also consider mentioned less conventional potential applications like attaining co-benefits from the CBD (e.g. Soto-Navarro et al. 2020).

Line 91 – The authority you seem to give the Bar-On estimate seems odd given that your maps could theoretically be used to improve their estimate. I would suggest (though, I do not require) that you change the language here to imply that Bar-On's estimate is speculative. Something like "Previous estimates have suggested that....". I would also explicitly describe the Bar-On estimate as a *carbon* stock estimate. By contrast, you map biomass (i.e. dry mass). If that distinction isn't made clear, readers may mistakenly compare your estimate to that of Bar-On which would not be inaccurate. Finally, I suggest that you report the Bar-On estimate in units of PgC (like you do later in the paper when presenting your own results) instead of GtC.

Line 94 – The body of this paragraph is a bit discordant with its first sentence. I think

your aim here is to say that biomass estimates are uncertain. If that's correct, I think you could do so more effectively. You might do so by changing the second and third sentences to something like: "However, our knowledge of the terrestrial biomass stock is relatively uncertain (Houghton et al. 2009). This uncertainty is well illustrated by the variance among forest biomass estimates: Pan et al..." Otherwise the transition from discussing total biomass C stocks (e.g. Bar-On and Houghton) to forest biomass C stocks (e.g. Pan, FRA) is an abrupt logical leap.

Line 105 – "Observable": do you mean "variable"? This is a term I haven't seen before.

Line 105 – "AGB": From this point forward, I think you're using AGB to implicitly refer to woody AGB (or the biomass of trees). If so, It's worth making that explicitly here. Many less-knowledgeable users of these maps mis-interpret them as representing the AGB of all plants rather than just that of trees and that leads to underestimates. I recommend explicitly defining it here (or earlier). Something like: "AGB (which hereafter we use to refer to aboveground biomass exclusively in forest trees)". And/or changing subsequent mention of "AGB" to "forest AGB". Do you use a certain definition of for forest/woody biomass? E.g. the FAO definition, A certain height, Etc.? If so, it would be worth stating that somewhere (if not here).

Line 115 – Baccini et al. (2012) represents 2007 and Baccini et al. (2017) represents even more recent years.

Line 118 – The year of the Erb reference should be 2018

Line 180 – I think you specifically mean investigations targeting *direct* estimation.

Line 198 – The last sentence of this brief paragraph is awkward. Consider rephrasing.

Line 298 – Does your uncertainty layer represent the standard deviation, as you say here, or the standard error? Below it sounds like you're propagating standard errors and the GlobBiomass website lists the uncertainty layer as representing the standard error (https://globbiomass.org/wp-

content/uploads/GB_Maps/Globbiomass_global_dataset.html). Whichever it is, please make it abundantly clear here in the text and make sure your description is accurate and consistent throughout the manuscript (and beyond).

Line 391 – It may also be worth comparing to regional/national AGB maps? Users often want to know whether it is appropriate to use global products to answer local questions. Obviously such a comparison wouldn't be comprehensive but even just showing for a few areas how well your map agrees with local products, might give users a sense of appropriate confidence or caution. This could also instead be done in the discussion if there are studies that have already made these comparisons with your data. One short comment on this manuscript notes that there is at least one such comparison that has been done in the U.S. Perhaps there are others as well and thus no need to do an extra analysis?

Line 407 – Again, is "standard deviation" correct here?

Figure 2 – How would you explain the areas with ~100% uncertainty? You don't mention these in the text but I suggest that you do. Are these areas that underwent a land cover change (fire, forest clearing, etc.) c.2010? Or does the model just do a poor job predicting biomass in certain areas (e.g. those with sparse woody vegetation)? It would likely help users to know this. Also, in my examination of your data, uncertainty can exceed the mean estimate (i.e. CV > 100%). How would you explain this to users? Is this a sign that error is not normally distributed? Also, this possibility should be indicated in the figure, either by allowing the colour-bar to surpass 100% or by changing the max label to something like "100+".

Line 418 – It would be prudent to state explicitly here (or wherever you report your first result) that the unit you're using is Mg of *dry matter* rather than carbon. Users occasionally assume that biomass maps are reporting carbon stocks.

Line 469 – I feel like your downplaying the fact that your mapped estimates saturate at high values. This is important for users to know and should be stated more frankly than

saying "albeit more gently". Without euphemism: "you underestimate high biomass stocks". So, the highest value you give above (757 Mg/ha) for the U.S. Pacific Northwest is likely a gross underestimate? You could illustrate the degree to which that might be true by referencing a field measured estimate from that region. In general, I don't mean to imply that this is fatal issue (every biomass map seems to have this problem) but you should be as transparent about it as possible about it.

Line 476 – The filled circles nicely show the saturation effect.

Line 495 – The highest estimate you give above (757 Mg/ha) is from the *temperate* rainforest of the U.S. Pacific Northwest. That would seem to contradict this statement.

Line 503 – I suggest breaking this discussion of the uncertainty layer out into a separate subsection (even if brief). Hardly any biomass maps are accompanied by an error layer so this could be viewed as a real strength of yours. As I've noted above (in the context of Figure 2), I think there are simple ways you could elaborate on this a bit to help users understand what it represents and how to use it.

Line 540 – I believe the FRA provides separate estimates for planted vs. natural forests. Are you considering both together here? If so, please make that clear in the text. I'm not sure about the 2010 FRA, but the 2015 version also considers "woodlands" separately. Presumably all of these categories fall under the purview of your map?

Line 557 – Mainly? It seems like a substantial portion could be due to your underestimates of high biomass.

Line 642 – At the U.S. state level, Spawn et al (2020) appear to compare your maps (separate from their modifications) to the U.S. FIA and show good agreement (see their figure 9). They don't employ the CCI Land Cover map in their comparisons so seemingly support your hypothesis that the difference is explained by the CCI map and not your biomass predictions?

Line 654 – I think "illustrating" is more appropriate, here, than "indicating".

Line 684 – The statement about minimal biomass at high latitudes warrants a citation.

Line 749 – I think you're saying, here, that if you further account for the woody biomass in non-forest CCI land cover classes, the total estimate increases? Please clarify in the text.

Line 751 – Here you're saying that your maps show 600 Pg of *woody* biomass. This is the number readers might improperly compare to the Bar-On *carbon* estimate you note in your introduction section – I'd make it abundantly clear here that this is 600 Pg of *dry biomass*, not carbon to prevent any unjust comparisons with Bar-On or others. This may also be an opportune place to mention the maps by Spawn et al. (2020) which use your woody biomass and additional maps of non-woody biomass to generate a total AGB (and separate BGB) estimate. Users may want to know when when/why one is more appropriate than the other. Whether here or elsewhere (perhaps the last paragraph?), I'd suggest (though, don't require) you make appropriate recommendations. Spawn et al. may also help put your number in appropriate context (i.e. make it more comparable to Bar-On).

Line 773 – please add "woody" before "AGB".

Overall, fantastic work!

---

## Referee Comment (RC2) · Anonymous Referee #2 · 7 Jun 2021

**1. GENERAL COMMENTS**

The manuscript addresses a topic of immense importance for climate change research and national reporting to international environmental agreements, providing improved geospatially explicit estimates of the global carbon pool of forest ecosystems.

It describes the development of global above ground biomass and growing stock volume maps at 1 ha spatial resolution from, primarily, satellite-borne radar data, and includes a thorough accuracy assessment. A comparison with other global AGB maps is provided, as well as with country reports from the FAO Forest Resources Assessment.

The manuscript is well-written and generally easy to follow, with the authors making great effort to clarify and discuss both strengths and weaknesses of the dataset. Only the sections on the assessment of global forest biomass resources (3.4) and comparison of AGB map estimates (3.5) are long and rather heavy to ingest for the reader. Consolidating these to reduce size somewhat would improve readability.

The dataset is easily accessible for download and the geotiff format accommodates straight-forward use with standard GIS software.

2. SPECIFIC COMMENTS

Line 136: "...PALSAR acquired images in the Fine Beam (FB) mode with 25 m spatial resolution..." should be "...PALSAR acquired images in the Fine Beam Dual-polarisation (FBD) mode with 20 m spatial resolution...". Note: acquisitions were made at 20 m resolution for the FBD data (and 10 m for FBS).

136-138: "Image acquisition followed a predefined observation scenario with the aim of achieving spatially and temporally consistent large-scale observational datasets". Add reference to ALOS PALSAR acquisition strategy: Rosenqvist, A., Shimada, M and Watanabe, M. ALOS PALSAR: A pathfinder mission for global-scale monitoring of the environment. IEEE Transactions on Geoscience and Remote Sensing, Vol. 45, No. 11, pp 3307-3316, Nov. 2007. doi.org/10.1109/TGRS.2007.901027

139: "...yearly mosaics of the radar backscatter...". Mention also that the PALSAR mosaics have been subject to radiometric terrain correction and are provided as gamma-0.

161-172: Please add a sentence to clarify how the time difference between the GLAS data (2003-2009) and the other datasets (approx. 2010) was handled to avoid systematic bias

Section 2.2 (AGB estimation): Throughout section 2.2 (and a bit of 2.3) and in equations 1, 3, 7 and 8, sigma-0 is used to denote the radar backscatter, while the SAR data (or at least the L-band mosaics) are provided as gamma-0. This probably does not matter for the models but a brief statement to acknowledge this would be useful to add.

239: "hyper-temporal". How many ASAR scenes were typically available for any given location? The word hyper-temporal gives the impression that Sentinel-1 style temporal density was available, which was hardly the case for ASAR (over perhaps Europe). Perhaps "multi-temporal" would be a more appropriate description?

260: "...uncompensated topographic effects in the ALOS PALSAR mosaics...". How did you parameterise the uncompensated topographic effects to define the weighting factor?

Could you also add a sentence (here or elsewhere in the text) about the cause of these uncompensated effects? Was it due to a lack or, or insufficient, radiometric slope correction in the PALSAR mosaics?

285: Explain how p1 and p2 in Eq. 6 are derived. Do they vary regionally?

395-396: "...the dataset were converted... from AGB carbon units (MgC ha-1) to AGB (Table S5)". A bit confusing statement. (Is this perhaps a typo?) Is there a difference between "AGB carbon units" and "AGB"? Table S5 offers no clue what you mean here. Please clarify.

The dataset comparisons in Section 3.1 illustrate the great spatial detail of this dataset compared to earlier mapping efforts and the importance of using high resolution satellite data as basis for the mapping. Impressive.

3.3 (Spatial distribution of AGB) For clarity, suggest to indicate early in the section that the acronyms used (TAr, TeDo, TeM, etc) refer to the FAO global ecological zone nomenclature (it becomes clear further down in Fig 6 but one is left wondering for a while)

3.4 (Assessment of global forest biomass resources). 3.5 (Comparison of AGB map estimates) The sections go into a high level of detail and are rather heavy to ingest for the reader. While I leave it to the discretion of the authors, consolidating these to reduce size somewhat would improve readability.

Figure 7: - The figure does not seem to indicate "Country AGB", but actually "Country AGB density", or rather "Country AGB average density", as you indicate the unit [MG/ha]. - Spent quite some time trying to get my head around the rather complex figure 7. It would be helpful to include some kind of legend which clarifies the different parameters you are using to illustrate not only a comparison between your AGB estimates at those of the FRA report, but also country absolute and relative forest area. Something like: - Size of circle - country forest area in CCI map (same scaling across all continents) - Colour of circle - relative country forest area on continent - Colour ramp - Add units to indicate (blue - smallest area on continent, brown - largest area on continent)

652: "All datasets showed...". Suggest to remind the reader about which datasets, AGB maps, you are referring to here (as one may have lost track of that after the rather dense section 3.4!)

Figure 10. Very helpful summary of section 3.5

722: "Overall, our maps reproduced...". Suggest to modify to "Overall, the results indicate that our maps reproduced..."

3. TECHNICAL CORRECTIONS

Line 129: "Phase" should be "Phased"

140: "publically" should be "publicly"

141: "0.00022° " should be "0.000225° "

809: "SoloEO" should be "soloEO" ;-)

Table S7. (very informative) "COUNTRY" should be "COUNTRY/TERRITORY" "{malvinas)" should be capitalised: "(Malvinas)" "Guinea0Bissau" Typo. Should be "Guinea Bissau" Taiwan is missing Suggest to report separately as "Republic of China (Taiwan)" and "People's Republic of China", resp. "The former Yugoslav Republic of Macedonia" new official name "North Macedonia"

---

## Author Response (AR1)

Dear Editor

We approach ESSD with a revised version of the manuscript "The global forest above-ground biomass pool for 2010 estimated from high-resolution satellite observations". The revision considered Short Comments and Reviewers Comments. We have already replied to the Short Comments individually. Our replies to the two reviews are part of this document. Our point-by-point replies are in italic. We hope that we have correctly addressed in our replies the issues raised in the discussion and in the review phase.

Best regards
Maurizio Santoro (on behalf of all co-authors)

**Replies to Reviewer Comments (RC1)**

Santoro et al eloquently describe their global maps of [woody] forest biomass in a way that is both thorough and approachable. Unlike previous biomass maps, they leverage global SAR composites to first generate a map of growing stock volume, from which they then generate an additional map of the biomass density. As I understand it, this approach contrasts with those used to generate previous forest biomass maps which have instead used more direct approaches that are thus more correlative and perhaps, then, less reliable. Santoro and colleagues also generate an uncertainty map, making their product distinct from its predecessors in that regard; I particularly appreciate the transparency this affords the product. Overall, I commend the authors for a well written paper and robust product. I am admittedly not an expert on non-optical remote sensing and would thus characterize my feedback below as more representative of a potential end user. While I can't speak much to the methods Santoro et al use, I do feel this perspective is important given the number and breadth of biomass maps that have been published in the last 10+ years. Below, I have identified several minor issues that I feel should be addressed to ensure that users like myself properly understand and interpret these data. Some are inevitably a matter of opinion and I respect that the authors may feel differently. Overall, though, I think this manuscript and the data it described will (at last!) be great contributions to our field.

*We are pleased that the manuscript has been well received. It is important that the dataset is correctly presented and we have taken care of implementing the changes suggested in this review. Our point by point reply to the comments below are in italic.*

Minor Comments:

Line 87 – Baccini et al. (2017) would seem to be an appropriate reference for forest degradation.

*Suggestion accepted.*

Line 85 – Gibbs et al. (2007) would seem to be an appropriate reference for REDD.

*It is indeed. The reference has been inserted.*

Line 86 – You might also consider mentioned less conventional potential applications like attaining co-benefits from the CBD (e.g. Soto-Navarro et al. 2020).

*Good point. Also this reference has been added.*

Line 91 – The authority you seem to give the Bar-On estimate seems odd given that your maps could theoretically be used to improve their estimate. I would suggest (though, I do not require) that you change the language here to imply that Bar-On's estimate is speculative. Something like "Previous estimates have suggested that....". I would also explicitly describe the Bar-On estimate as a *carbon* stock estimate. By contrast, you map biomass (i.e. dry mass). If that distinction isn't made clear, readers may mistakenly compare your estimate to that of Bar-On which would not be inaccurate. Finally, I suggest that you report the Bar-On estimate in units of PgC (like you do later in the paper when presenting your own results) instead of GtC.

*These are valuable suggestions and have been implemented. We agree that the Bar-on numbers are based on less accurate data than the dataset here presented. The reason for citing Bar-on's work was primarily to highlight that most of the live biomass on Earth is stored in woody vegetation.*

Line 94 – The body of this paragraph is a bit discordant with its first sentence. I think your aim here is to say that biomass estimates are uncertain. If that's correct, I think you could do so more effectively. You might do so by changing the second and third sentences to something like: "However, our knowledge of the terrestrial biomass stock is relatively uncertain (Houghton et al. 2009). This uncertainty is well illustrated by the variance among forest biomass estimates: Pan et al..." Otherwise the transition from discussing total biomass C stocks (e.g. Bar-On and Houghton) to forest biomass C stocks (e.g. Pan, FRA) is an abrupt logical leap.

*That's correct. We have modified the text following these suggestions while also trying to keep the original meaning of the paragraph, with Bar-on's work being cited primarily for their numbers on biomass repartition.*

Line 105 – "Observable": do you mean "variable"? This is a term I haven't seen before.

*We have slightly rephrased this sentence and focus now on the concept of "measurement".*

Line 105 – "AGB": From this point forward, I think you're using AGB to implicitly refer to woody AGB (or the biomass of trees). If so, It's worth making that explicitly here. Many less-knowledgeable users of these maps mis-interpret them as representing the AGB of all plants rather than just that of trees and that leads to underestimates. I recommend explicitly defining it here (or earlier). Something like: "AGB (which hereafter we use to refer to aboveground biomass exclusively in forest trees)". And/or changing subsequent mention of "AGB" to "forest AGB". Do you use a certain definition of for forest/woody biomass? E.g. the FAO definition, A certain height, Etc.? If so, it would be worth stating that somewhere (if not here).

*The paragraph between lines 104 and 114 is a review of previous mapping activities on "AGB" from a method perspective. Each activity might have had its own definition of AGB, which is, however, in this context of minor relevance. Further down, we review AGB data products; at this stage, we now provide details on the type of AGB that was estimated also taking into account that these data products have been assessed in our study. Our definition of AGB was provided first on line 119 (forest AGB); following this comment, the definition has been updated in the revision as well. The detailed definition of the AGB mapped in the dataset is provided in Section 3.1. Overall, we have revised the text to ensure that we always refer to forest AGB and woody biomass.*

Line 115 – Baccini et al. (2012) represents 2007 and Baccini et al. (2017) represents even more recent years.

*We have revised the sentence by referring to the full range of years covered by all global and biome-specific datasets cited two lines above. Baccini et al., 2017, is not referred to in this context because not publicly available and because not contributing to a global dataset.*

Line 118 – The year of the Erb reference should be 2018

*Thanks for spotting this. The reference has been updated.*

Line 180 – I think you specifically mean investigations targeting \*direct\* estimation.

*Yes. We have modified this sentence according to this comment.*

Line 198 – The last sentence of this brief paragraph is awkward. Consider rephrasing.

*We have removed the second part of the sentence and linked the first part with the text in the next sentence.*

Line 298 – Does your uncertainty layer represent the standard deviation, as you say here, or the standard error? Below it sounds like you're propagating standard errors and the GlobBiomass website lists the uncertainty layer as representing the standard error (https://globbiomass.org/wp-content/uploads/GB_Maps/Globbiomass_global_dataset.html). Whichever it is, please make it abundantly clear here in the text and make sure your description is accurate and consistent throughout the manuscript (and beyond).

*It is standard deviation. The term "standard error" was erroneously used in initial reporting. We ensure that the data distribution also refers to standard deviation.*

Line 391 – It may also be worth comparing to regional/national AGB maps? Users often want to know whether it is appropriate to use global products to answer local questions. Obviously such a comparison wouldn't be comprehensive but even just showing for a few areas how well your map agrees with local products, might give users a sense of appropriate confidence or caution. This could also instead be done in the discussion if there are studies that have already made these comparisons with your data. One short comment on this manuscript notes that there is at least one such comparison that has been done in the U.S. Perhaps there are others as well and thus no need to do an extra analysis?

*It is an obvious question to which extent this paper should compare the global dataset with local maps. This work was actually undertaken but the results were not easy to interpret. Discrepancies were often related to the different methods used to estimate biomass from the remote sensing data. Given the already long and dense paper in its current format, we opted to omit such analyses. Nevertheless, the suggestion is valuable and we have added a sentence to explain why we restricted our analysis to "global" maps. We have also added reference to the first few papers that have investigated the local value of the dataset. Considering the still small number of scientific publications highlighting the usefulness and caveats of the dataset, it is preferred to stay with the citations without attempting to provide a big picture, to which extent such a global dataset can be of use at the local scale.*

Line 407 – Again, is "standard deviation" correct here?

*Standard deviation is correct.*

Figure 2 – How would you explain the areas with ~100% uncertainty? You don't mention these in the text but I suggest that you do. Are these areas that underwent a land cover change (fire, forest clearing, etc.) c.2010? Or does the model just do a poor job predicting biomass in certain areas (e.g. those with sparse woody vegetation)? It would likely help users to know this. Also, in my examination of your data, uncertainty can exceed the mean estimate (i.e. CV > 100%). How would you explain this to users? Is this a sign that error is not normally distributed? Also, this possibility should be indicated in the figure, either by allowing the colour-bar to surpass 100% or by changing the max label to something like "100+".

*This comment made us aware that we omitted mentioning that the colour bars in Figure 2 were truncated at 500 Mg ha$^{-1}$ AGB and 100% uncertainty to increase contrast. This information has been added to the figure caption. We have also updated the legend of each colour bar with a "+" in correspondence of the upper bound. A paragraph describing the map of the relative standard deviation in Fig. 2 has been added in reply to this comment.*

*"The AGB standard deviation, expressed in Fig. 2 relative to the AGB estimates, was on average 50% with an inter-quartile range of values between 44% und 61%. The rather constant relative uncertainty is also illustrated by the horizontal bars in the latitudinal profile of Fig. 2, which scale with the AGB level. The relative standard deviation was smaller than 100% for approximately 95% of the mapped pixels which explains truncating the colour bar of the AGB standard deviation map in Fig. 2 at 100%. The large majority of the AGB estimates for which the standard deviation exceeded the 100% level were below 20 Mg ha-1 such as in sparsely vegetated regions corresponding to the transition to tundra in Canada and Alaska, the Siberian Lowlands and Far East Russia or in poorly stocked forests such as in northern China and west Madagascar." (Lines 442-448)*

Line 418 – It would be prudent to state explicitly here (or wherever you report your first result) that the unit you're using is Mg of *dry matter* rather than carbon. Users occasionally assume that biomass maps are reporting carbon stocks.

*This is a valuable comment and we share this view. When defining our data product (Introduction), we refer to the unit in terms of Mg of dry matter per hectare.*

Line 469 – I feel like your downplaying the fact that your mapped estimates saturate at high values. This is important for users to know and should be stated more frankly than saying "albeit more gently". Without euphemism: "you underestimate high biomass stocks". So, the highest value you give above (757 Mg/ha) for the U.S. Pacific Northwest is likely a gross underestimate? You could illustrate the degree to which that might be true by referencing a field measured estimate from that region. In general, I don't mean to imply that this is fatal issue (every biomass map seems to have this problem) but you should be as transparent about it as possible about it.

*We admit that the wording used to support the interpretation of the scatter plots in Figure 5 (original manuscript) was a delicate issue. When looking at the global scatter plot, we agree that there is a strong underestimation at high AGB. The question is whether this is a systematic issue, i.e., it occurred everywhere, or a local issue. For this reason, we split the data per continent. These scatter plots reveal that the under-estimation differs depending on the continent. More to that, the largest underestimation occurred in Madagascar and Tasmania (reported in Table S8 and Figures S12, original manuscript).These islands harbor some of the densest forests on Earth but in terms of magnitude, the carbon stocks in these two regions are not comparable to the stocks from all other regions for which reference data was available and for which the scatter plots revealed minor over/underestimation. The "albeit more gently" tried to summarize these multiple evidences and we believe that it is a correct way of reporting our interpretation of the scatter plot.*

*Following this comment, we seem to understand that readers may misinterpret our assessment by inferring that if AGB is underestimated starting at 250 Mg/ha, the higher the AGB the larger the underestimate. To avoid this interpretation, we displayed in Figure S12 (original manuscript) an assessment of averages at the level of administrative units. The patterns differ depending on the region but there is no such evidence of a constant "saturation" of the averages or an increase of underestimation, which further confirms the indication by the continental scatter plots. Our strategy concerning validation, verification and assessment of the AGB map is outlined in Section 2.4.*

*With respect to the specific AGB of 757 Mg/ha as for any single estimate of AGB, we are not in the position to conclude whether this value is positively or negatively biased. This aspect is touched upon in the manuscript and we would refer to Herold et al. (2019) for a wider perspective on validation of maps.*

Line 476 – The filled circles nicely show the saturation effect.

*We would not refer to it as saturation when looking at the continental scatter plots (see also our reply to the previous comment) but indeed the reason for including filled circles was to aid the interpretation of the validation results.*

Line 495 – The highest estimate you give above (757 Mg/ha) is from the \*temperate\* rainforest of the U.S. Pacific Northwest. That would seem to contradict this statement.

*The sentence has been rephrased to avoid this contradicting statement.*

Line 503 – I suggest breaking this discussion of the uncertainty layer out into a separate subsection (even if brief). Hardly any biomass maps are accompanied by an error layer so this could be viewed as a real strength of yours. As I've noted above (in the context of Figure 2), I think there are simple ways you could elaborate on this a bit to help users understand what it represents and how to use it.

*Thanks for the sign of appreciation. The idea was to present the AGB map and the uncertainty layer (Fig. 2, original manuscript) with some related analyses (Fig. 6, original manuscript) one after the other. By splitting the presentation of the uncertainty layer from the AGB layer, we fear that there would be some back and forth flipping when reading the paper. We value the suggestion but prefer to keep the structure of the results unaltered. Nevertheless, we have added a description of the AGB standard deviation (in addition to the text written to accompany Fig. 2, see comment above) and some explanations concerning the origin of the large uncertainty in some regions. In support of this explanation, we have inserted a new figure in the Supplement (Fig S11). We hope that this approach is satisfactory.*

Line 540 – I believe the FRA provides separate estimates for planted vs. natural forests. Are you considering both together here? If so, please make that clear in the text. I'm not sure about the 2010 FRA, but the 2015 version also considers "woodlands" separately. Presumably all of these categories fall under the purview of your map?

*This is a well-thought comment. We spent a considerable amount of time to ensure that we would compare the right numbers. FRA 2010 provided separate AGB numbers for forest land and other wooded land. It also provided area statistics for forest land, tree out of forests and other land. Here, we used the numbers reported by each country for forest area and forest AGB. In terms of AGB, the OWL AGB is seldom reported. When reported, it represented a minor fraction of the total (forest+OWL) AGB except for a very small number of countries in Africa and Asia. Finally, accounting for OWL AGB in our map-to-FRA assessment was negligible because the density and stocks were scaled accordingly with the area of the tree out of forest category, representing the extent of OWL.*

Line 557 – Mainly? It seems like a substantial portion could be due to your underestimates of high biomass.

*We are not sure that we understood this comment correctly. While we estimate 17% more AGB than the FRA and our estimate of the forest area is 40% larger than in the FRA, it makes sense that the AGB stock is 64% larger.*

Line 642 – At the U.S. state level, Spawn et al (2020) appear to compare your maps (separate from their modifications) to the U.S. FIA and show good agreement (see their figure 9). They don't employ the CCI Land Cover map in their comparisons so seemingly support your hypothesis that the difference is explained by the CCI map and not your biomass predictions?

*It is important that an independent study came to similar conclusions. As a side note, in a separate study, we analyzed the area attributed by CCI to forest in the U.S. and noticed some remarkable differences with respect to the state-wise FIA values. When recalibrating the AGB stocks per state with the ration (forest area CCI / forest area FIA), we obtain a very good agreement with the stocks estimated by FIA.*

Line 654 – I think "illustrating" is more appropriate, here, than "indicating".

*Corrected.*

Line 684 – The statement about minimal biomass at high latitudes warrants a citation.

*We emphasize that the 30 Mg/ha level was not exceeded by any of the datasets. To confirm the validity of this AGB level, we introduced citations based on published data from inventory measurements.*

Line 749 – I think you're saying, here, that if you further account for the woody biomass in non-forest CCI land cover classes, the total estimate increases? Please clarify in the text.

*This is correct. The text has been amended.*

Line 751 – Here you're saying that your maps show 600 Pg of *woody* biomass. This is the number readers might improperly compare to the Bar-On *carbon* estimate you note in your introduction section – I'd make it abundantly clear here that this is 600 Pg of *dry biomass*, not carbon to prevent any unjust comparisons with Bar-On or others. This may also be an opportune place to mention the maps by Spawn et al. (2020) which use your woody biomass and additional maps of non-woody biomass to generate a total AGB (and separate BGB) estimate. Users may want to know when when/why one is more appropriate than the other. Whether here or elsewhere (perhaps the last paragraph?), I'd suggest (though, don't require) you make appropriate recommendations. Spawn et al. may also help put your number in appropriate context (i.e. make it more comparable to Bar-On).

*We have clarified that our AGB estimates are relative to biomass and added a number for carbon so to make numbers comparable in this unit. We also added a reference to Spawn et al. to highlight how the integration of our dataset with other can provide a more comprehensive picture of the global carbon stocks in vegetation. Nonetheless, we prefer to avoid a comparison of numbers from other studies that is beyond our analysis (lines 749-760 of the original manuscript) because it would involve an explanation of discrepancies. Here, the nuances of definitions adopted by the different study may be easily overlooked, leading to a potentially wrong interpretation of numbers.*

Line 773 – please add "woody" before "AGB".

*Added.*

Overall, fantastic work!

*Thanks*

**Replies to Reviewer Comments (RC2)**

1. GENERAL COMMENTS
The manuscript addresses a topic of immense importance for climate change research and national reporting to international environmental agreements, providing improved geospatially explicit estimates of the global carbon pool of forest ecosystems. It describes the development of global above ground biomass and growing stock volume maps at 1 ha spatial resolution from, primarily, satellite-borne radar data, and includes a thorough accuracy assessment. A comparison with other global AGB maps is provided, as well as with country reports from the FAO Forest Resources Assessment. The manuscript is well-written and generally easy to follow, with the authors making great effort to clarify and discuss both strengths and weaknesses of the dataset. Only the sections on the assessment of global forest biomass resources (3.4) and comparison of AGB map estimates (3.5) are long and rather heavy to ingest for the reader. Consolidating these to reduce size somewhat would improve readability. The dataset is easily accessible for download and the geotiff format accommodates straight-forward use with standard GIS software.

*We are thankful for the review, the comments to the manuscript and the positive feedback to the data product. We have implemented the suggested modifications. We have also thought how to reduce the amount of information in Sections 3.4 and 3.5. Both sections underwent substantial pruning during the internal review before the original manuscript was submitted. In all honesty, we find it hard to further consolidate these Sections without impacting the overall message given by this paper in terms of reliability and new evidences. To better guide readers through Section 3.4, we have split results in two sub-sections. Some modifications were introduced in Section 3.5 as well but without changing its structure. Our point by point reply to the comments below are in italic.*

2. SPECIFIC COMMENTS
Line 136: ": : :PALSAR acquired images in the Fine Beam (FB) mode with 25 m spatial resolution: : :" should be ": : :PALSAR acquired images in the Fine Beam Dual polarisation (FBD) mode with 20 m spatial resolution: : :". Note: acquisitions were made at 20 m resolution for the FBD data (and 10 m for FBS).

*Corrected.*

136-138: "Image acquisition followed a predefined observation scenario with the aim of achieving spatially and temporally consistent large-scale observational datasets". Add reference to ALOS PALSAR acquisition strategy: Rosenqvist, A., Shimada, M and Watanabe, M. ALOS PALSAR: A pathfinder mission for global-scale monitoring of the environment. IEEE Transactions on Geoscience and Remote Sensing, Vol. 45, No. 11, pp 3307-3316, Nov. 2007. doi.org/10.1109/TGRS.2007.901027139: ": : :yearly mosaics of the radar backscatter: : :". Mention also that the PALSAR mosaics have been subject to radiometric terrain correction and are provided as gamma-0.

*One sentence and a reference have been added.*

161-172: Please add a sentence to clarify how the time difference between the GLAS data (2003-2009) and the other datasets (approx. 2010) was handled to avoid systematic bias.

*We have added one sentence to explain that the LiDAR observations were not used as predictors of GSV but to derive estimate of model parameters that can be considered time invariant. For this reason, the impact of having a LiDAR dataset not contemporary to the SAR observations is minimal.*

Section 2.2 (AGB estimation): Throughout section 2.2 (and a bit of 2.3) and in equations 1, 3, 7 and 8, sigma-0 is used to denote the radar backscatter, while the SAR data (or at least the L-band mosaics) are provided as gamma-0. This probably does not matter for the models but a brief statement to acknowledge this would be useful to add.

*The symbols with sigmas in these Sections and Equations indeed need to be interpreted in a generic way as backscattering coefficients. To make sure that in particular the symbol sigma0_for is correctly interpreted, we now explain that it represents "the forest backscatter, i.e., the SAR backscatter observation from an ALOS PALSAR or an Envisat ASAR image (Section 2.1)" directly after Eq. (1) is introduced. We hope that this is sufficient without having the need to specify that an image was processed to a certain type of backscatter coefficient.*

239: "hyper-temporal". How many ASAR scenes were typically available for any given location? The word hyper-temporal gives the impression that Sentinel-1 style temporal density was available, which was hardly the case for ASAR (over perhaps Europe). Perhaps "multi-temporal" would be a more appropriate description?

*Thank you for this comment. The term hyper-temporal was first used in relation with previous studies with ASAR observations of the northern hemisphere to indicate a very large number of observations (> 100). In support of our statement on the "hyper-temporal dataset", we have realized that we did not present a figure showing the number of observations available per pixel. In the revision, the term "hyper-temporal" has been removed because redundant. At the same time, we have introduced a figure in the Supplement (Fig. S1) showing the number of observations, which by itself is in our opinion important to understand the necessity of a large number of observations at C-band to ensure that the resulting estimates of GSV are reliable.*

260: ": : :uncompensated topographic effects in the ALOS PALSAR mosaics: : :". How did you parameterise the uncompensated topographic effects to define the weighting factor? Could you also add a sentence (here or elsewhere in the text) about the cause of these uncompensated effects? Was it due to a lack or, or insufficient, radiometric slope correction in the PALSAR mosaics?

*Since we decided to focus on the data product in our manuscript, technical aspects not directly involved in the evaluation of the results were reported in the Supplement. In this case, we refer to Eq. (S10) in Section A.4 of the Supplement. However, in our revision we have added "residual" to "uncompensated topographic effects" in the main body of the manuscript to clarify that topography was compensated but because of either the quality of the DEM or the procedure adopted, the correction did not completely compensate for the effect of sloped terrain on the backscattered intensity.*

285: Explain how p1 and p2 in Eq. 6 are derived. Do they vary regionally?

*Again, since technical aspects are reported in the Supplement, we refer to Section A.5 in the Supplement as well as Quegan et al., 2017. Both are duly cited in the main body of the manuscript. In reply to the question, in situ data on stem biomass and total biomass were stratified by biome and the model was fitted for each biome. Unfortunately, the paucity of data did not allow for a finer regionalization. The implications are discussed throughout the paper at several instances.*

395-396: ": : :the dataset were converted: : : from AGB carbon units (MgC ha-1) to AGB (Table S5)". A bit confusing statement. (Is this perhaps a typo?) Is there a difference between "AGB carbon units" and "AGB"? Table S5 offers no clue what you mean here. Please clarify. The dataset comparisons in Section 3.1 illustrate the great spatial detail of this dataset compared to earlier mapping efforts and the importance of using high resolution satellite data as basis for the mapping. Impressive.

*AGB refers to the density of the organic mass, AGBC refers to the density of the carbon mass. AGBC is roughly half of the AGB. We have rewritten the sentence to avoid misunderstanding by specifying the carbon fraction value used to convert AGBC to AGB.*

3.3 (Spatial distribution of AGB) For clarity, suggest to indicate early in the section that the acronyms used (TAr, TeDo, TeM, etc) refer to the FAO global ecological zone nomenclature (it becomes clear further down in Fig 6 but one is left wondering for a while)

*To avoid this issue, we cite now Fig. 6 before any of the FAO acronyms are used in the text.*

3.4 (Assessment of global forest biomass resources). 3.5 (Comparison of AGB map estimates) The sections go into a high level of detail and are rather heavy to ingest for the reader. While I leave it to the discretion of the authors, consolidating these to reduce size somewhat would improve readability.

*We agree that Section 3.4 is very dense and long. This Section underwent substantial pruning before submission and reports the main results at global/continental and national level. In our opinion, it is important to report result as such level of detail to build confidence in the dataset. Also, our paper targets a large variety of readers with interests that range from global perspective to national interests. To account for different points of view, we have broken down the Section into two sub sections, each dealing with one of the levels mentioned above. We hope that this minimal intervention can be considered acceptable.*

*Section 3.5 is also dense and it underwent pruning before submission. After reading this Section through, we have identified several reasons for opting to leave it unchanged. The text is not particularly long for a results section, as it extends over 1 and ½ page and the is supported by two figures and one table. Grouping the Section according to an analysis per biome would have resulted in many small fragments. As for the assessment against the FRA data, we wanted to ensure that the comparison of maps is done and reported thoroughly to provide evidences concerning the reliability of our map product in a broad context. Finally, we think that this Section attracts the attention of a readership interested in map-based estimates of biomass, who probably pays less attention to the content of Section 3.4.*

Figure 7: - The figure does not seem to indicate "Country AGB", but actually "Country AGB density", or rather "Country AGB average density", as you indicate the unit [MG/ha]. - Spent quite some time trying to get my head around the rather complex figure 7. It would be helpful to include some kind of legend which clarifies the different parameters you are using to illustrate not only a comparison between your AGB estimates at those of the FRA report, but also country absolute and relative forest area. Something like: - Size of circle - country forest area in CCI map (same scaling across all continents) - Colour of circle - relative country forest area on continent – Colour ramp - Add units to indicate (blue - smallest area on continent, brown - largest area on continent)

*Throughout Section 3.4, which includes Figure 7, we have noticed this misleading wording (country AGB) and have replaced it with average AGB. The suggestions to improve the readability of Figure 7 are very valuable. The legend of the color ramp has been added. The figure caption has been modified.*

652: "All datasets showed: : :". Suggest to remind the reader about which datasets, AGB maps, you are referring to here (as one may have lost track of that after the rather dense section 3.4!)

*We refer now to the "map datasets" and to Table 3 where the datasets are listed.*

Figure 10. Very helpful summary of section 3.5

*Thanks for the positive feedback*

722: "Overall, our maps reproduced: : :". Suggest to modify to "Overall, the results indicate that our maps reproduced: : :"

*Corrected*

3. TECHNICAL CORRECTIONS
Line 129: "Phase" should be "Phased"
140: "publically" should be "publicly"
141: "0.00022_ " should be "0.000225_ "
809: "SoloEO" should be "soloEO" ;-)

*All corrected.*

Table S7. (very informative) "COUNTRY" should be "COUNTRY/TERRITORY" "{malvinas)" should be capitalised: "(Malvinas)" "Guinea0Bissau" Typo. Should be "Guinea Bissau" Taiwan is missing Suggest to report separately as "Republic of China (Taiwan)" and "People's Republic of China", resp. "The former Yugoslav Republic of Macedonia" new official name "North Macedonia"

*We have updated the table following the terminology used in the FRA. "Country" has been replaced with "Country/area". There is no country report for the Republic of China (Taiwan) nor there is an entry in the FRA 2010 global tables . For North Macedonia we used the name adopted by the country in the country report submitted to the FRA 2010. All other typos have been corrected.*

---

## Author Response (AR2)

Dear Editor

We approach ESSD with a newly revised version of the manuscript "The global forest above-ground biomass pool for 2010 estimated from high-resolution satellite observations". Following the comments by RC #3. Our replies are part of this document. Our point-by-point replies are in italic.

Best regards
Maurizio Santoro (on behalf of all co-authors)

Only a few minor editing suggestions but otherwise, this is a comprehensive and ground-breaking study that has led to the generation of global maps of above ground biomass from radar data for 2010. The technique focuses on the retrieval of GSV and subsequently AGB (by using information on wood density and expansion factors) without the need for ground calibration but the validation is robust/rigorous and uses a substantive dataset. The text clearly and concisely explains the approach and deals with a range of important issues, including comparisons with existing ground and satellite-based efforts, the amounts and distribution of biomass globally and within regions and relevance to global and country level reporting of biomass (vegetation carbon stocks).

Line 66: Should read "the literature"
Line 782: , such as savannah,
Line 736: Comma after 90N)

*We are thankful for the positive comments. We have implemented all the changes suggested.*